# Traditional health practices: A qualitative inquiry among traditional health practitioners in northern Uganda on becoming a healer, perceived causes of illnesses, and diagnostic approaches

Amos Deogratius Mwaka[1,2]*, Jennifer Achan[3], Christopher Garimoi Orach[3,4]

1 Department of Medicine, School of Medicine, College of Health Sciences, Makerere University, Kampala, Uganda, 2 Faculty of Medicine, Department of Medicine, Gulu University, Gulu, Uganda, 3 School of Public Health, College of Health Sciences, Msakerere University, Kampala, Uganda, 4 Department of Community Health & Behavioral Sciences, School of Public Health, College of Health Sciences, Makerere University, Kampala, Uganda

* mgratius@gmail.com

**Data Availability Statement:** All relevant data are within the manuscript.

## Abstract

### Background

The practice of traditional and complementary medicine is increasing in most low-and-middle-income countries especially for chronic communicable and non-communicable diseases. In this study, we aimed to understand how people gain healing power and become traditional health practitioners (THPs), perceived causes of illnesses, and how THPs diagnose illnesses.

### Methods

This was a cross-sectional qualitative interview-based study. We used semi-structured in-depth guides to collect data from THPs identified through the Acoli cultural institutions and snowball sampling technique. The study team visited the THPs and interviewed them in their homes. Some THPs allowed the study team to visit them more than once and as well observe their healing practices and medicines. Thematic analysis approach was used to analyze the data. Atlas.ti version 9.2 was used to support data analysis.

### Results

Twenty two THPs aged 39–80 years were included in the study. Most of the respondents were male, and married. We identified three main themes: (i) how people gain healing power and become traditional health practitioners; (ii) perceived causes of illnesses; and (iii) how illnesses are diagnosed. The majority of respondents reported that most people become THPs through: inheriting healing power from their parents or grandparents; transfer of healing powers from senior healers; instructions during visions and dreams; and, acquiring healing power during spirits possessions. Perceived causes of illnesses included: fate

**Funding:** ADM: Alborada Cambridge Africa Research Fund 2017 The funders had no role in study design, data collection and analysis, decision to publish, or preparation of the manuscript.

**Competing interests:** The authors have declared that no competing interests exist.

**Abbreviations:** GUREC, Gulu University Research and Ethics Committee; HC, Health Centre; HIC, High Income Countries; IDP, Internally Displaced Persons; LMIC, Low and Middle–Income Countries; LRA, Lord's Resistance Army; NRA, National Resistance Army; NRM, National Resistance Movement; THP, Traditional Health Practitioners; T&CM, Traditional and Complementary Medicine.

and natural causes, spirits attacks, curses by elders, witchcraft, contagion and infections, poor hygiene, heredity, and malevolent actions. THPs diagnose illnesses through various approaches including consultations with spirits, observing patterns of occurrences and events, evaluation of symptoms and signs of illnesses, use of bones from animals/birds and other objects to diagnose illnesses, performing diagnostic rituals, and using biomedical laboratory testing in health facilities.

## Conclusion

Healing knowledge and powers are acquired in particular ways that can be traced to appraise authenticity of healers during registration and licensing to ensure safety of patients. Understanding perspectives of the THPS on causes of illnesses and how diagnoses are made potentially informs strategies for integration and or collaboration between the national biomedical health system and traditional health practices.

## Introduction

The use of traditional and complementary medicines (T&CMs) has increased in recent decades in both the high income countries (HICs) and the low- and middle-income countries (LMICs). Although the use of T&CM is apparently more common in low- and middle-income countries (LMICs) [Africa; 40% (95%CI: 23–58) and Asia; 28%; (95%CI: 21–35)], the use of these remedies are also still common in the high income countries (HICs) [17% (95%CI: 14–21)] [1]. For example, the use of T&CM among indigenous population across the USA, Australia, New Zealand and Canada ranged between 19% and 57.7% [2]. In particular, 13% to 56% of cancer patients from high income countries in Western and Northern Europe, USA and Canada use T&CM and or alternative therapies [3–8]. In Asia, 13% to 61% of cancer patients use T&CM [9–11]. In low-income and lower middle-income countries, especially in sub-Saharan Africa, 65% to 79% of cancer patients use T&CM [12]. About 60% of cancer patients in sub Saharan Africa use T&CM concurrently with conventional cancer therapies [13]. Some evidence shows that use of T&CM potentially delays health seeking at biomedical facilities (e.g., hospitals, clinics). For example, a study of 258 people living with HIV/AIDS at different stages of illness, 48 family members of people who died from HIV, and 53 HIV healthcare workers, revealed that use of T&CM was quite common and potentially led to delays in seeking biomedical care [14]. In most societies in the LMICs, traditional health practitioners (THPs) are popular among people because they overcome deficiencies in biomedicine with regards to cultural beliefs of patients and their families including the perceived causes of illnesses [15, 16]. People listen to THPs, respect them and take what the THPs say about their health and illnesses seriously [17]. Therefore, THPs have remained central in the healthcare of their people mainly because they understand the cultural values, language and behaviours of the people in their communities and are therefore well placed to meet the people's expectations in the process of care [18, 19].

Studies among the Acoli in northern Uganda revealed that use of T&CM is common and are used for a variety of reasons including limited access to biomedicine, long waiting times at health facilities, and absenteeism of healthcare professionals [20]. The Acoli are a part of the greater Luo ethnic group that are mainly found in northern and south eastern Uganda, southern South Sudan, north eastern Democratic Republic of Congo (DRC), north eastern

Tanzania, western Kenya, and Ethiopia. In addition, there are ethnic Acoli/Luo people in many other places in the world including New Zealand [21–24]. The life and civilization of the Acoli as the Central Luo may be divided into four epochs: the precolonial era, colonial and immediate post-independent era, the Lord's Resistance Army and National Resistance Army (LRA/NRA) conflict era, and the post-internally displaced person (IDP) camps era. The violent LRA/NRA war that affected the Central Luo spanned from 1987 to 2006, causing massive displacement of the people into the infamous internally displaced persons (IDP) camps, euphemistically referred to as protected villages. There has been serious degeneration of the Acoli cultural practices and norms including traditional health practices during the LRA/NRA war [25, 26]. Therefore, there is need to interrogate and understand the Acoli people especially their health practices post the devastating war. A study among the Luo of western Kenya revealed that the healing practices of the Luo people have been characterized by informal learning about traditional medicines and healing processes through apprenticeship from grandparents and parents. Learning occurs through participation in practical daily life experiences and social relations [27]. Luo children are often involved in domestic chores including cooking, digging in the gardens, fetching water, collecting firewood and harvesting crops. Parents and grandparents identify children who are interested in certain key activities and mentor them. For example, not all boys would necessarily be taken along in hunting expeditions, but only those who have demonstrated skills and interests in the activity. Similarly, parents and grandparents who are healers identify children and mentor them in healing practices, showing them medicinal plants and how to mix and administer them for different illnesses [27, 28]. The grandparents in Luo societies were sages who guided their families especially in times of uncertainties. They had a wealth of knowledge of the culture, arising from their long life experiences, older age, and links with the past [27]. In the Luo societies, people became healers through different ways; first, the deep knowledge required to become a THP is often gained through apprenticeship with grandparents and parents. Second, the healers receive guidance and often maintain close contact with the spirits of the elders who pass to them healing powers. Another way people became healers is through personal experience with illnesses in oneself or one's own child or relative, and successful trial of medicines. When such medicines tried have proved to consistently work for the particular illnesses, then one gains a reputation for being a healer of those particular illnesses [27]. Once a person has become known for successfully treating particular illnesses and has gained trust for the same, then people from beyond the extended families seek care with him/her. In Uganda, there is inadequate description of the growth and development of the Luo healing practices especially among the Acoli following the long period of internment of the population in the internally displaced person's (IDP) camps. The two decades LRA/NRM war used some of the established Judeo-Christian religious principles including the Ten Commandments to mobilize and encourage their troupes to bravery, as well as coerce the population to comply with their ideologies [29–31]. It is not well understood why the Holy Spirit Movement and LRA used Christian principles to enforce their mobilizations and campaigns. One clear fact though was that the LRA leadership based on these Christian principles to condemn and kill people who did not comply with their commands interpreted within the context of religion. They convinced the population that killing under such circumstances was justified and would therefore not raise hairs against the executioners [29–31]. The rebels strictly prohibited Acoli time-honored practices including divination (pejoratively referred to as witchcraft) and rewarded those who did not comply with severe punishments including death. Therefore, consultations with traditional healers, most of whom doubled as diviners were abandoned in the Acoli sub region. Yet, the Acoli were accustomed to diagnosis of their social and health problems as well as predicting their future through divination. The conflict therefore had potential to severely disintegrate the

culture of the Acoli people. In the words of Fr Carlos, "The Acoli fear they are losing vital aspects of their culture" [32]. Other scholars described the conflict as a well-planned genocide, intended to wipe the Acoli and her powerful culture [33]. The feared loss of culture and cultural values include traditional health practices which are not simply treatment of diseases, but rather a way of life of a people. Therefore, in this study, we aimed to describe the post conflict status of traditional health practices among the Central Luo. Specifically, we focused on how one gains healing powers, learns about medicines that treat particular illnesses and becomes a traditional health practitioner (THP), as well as healers' perceived causes of illnesses, and diagnostic approaches to illnesses in the context of the Acoli people.

## Materials and methods

### Study design

This was an exploratory qualitative study that used in-depth interviews and observations of the practices and memorabilia of the traditional healers. The qualitative approaches allow for deep explorations of the meanings, views, and experiences of respondents [34–36].

### Study setting

The study was conducted in eight districts in northern Uganda. The people in these eight districts are mainly Acoli, a Nilotic ethnic group that historically migrated from present day South Sudan [37, 38]. The people in the study region experienced violence and displacement into internally displaced persons (IDP) camps resulting from an armed conflict between the National Resistance Army (NRA) and the rebel group Lord's Resistance Army (LRA) from 1987 till 2006 [25].

The Uganda national health system is composed of the public sector, private-for-profit (PFP), and private not-for-profit (PNFP) providers. The public sector is graded from health center (HC) I to the National Referral Hospitals, depending on the level of services provided; Health Center I (HC I) provides mainly health promotion and preventive health services, while Health Center II (HC II) have physical infrastructures and provide outpatient curative services in addition to the services at HC I. Health Center III provide inpatient, laboratory, and maternal child health services in addition to the services provided at HC II. In addition to the services at HC III, Health Center IV (HC IV) provide emergency surgical and laboratory services. Level five is the district general hospitals that provide higher level services including radiology, laboratory and general surgical care. Secondary healthcare services are provided at the Regional Referral Hospitals (RFH) while tertiary and super-specialized care services are provided by the National Referral Hospitals [39].

In Uganda, traditional and complementary medicine (T&CM) practices have not been integrated into the national health system. However, there is a law that officially recognizes and regulates the use of Traditional and Complementary Medicine more or less as a parallel system to the national biomedical health system [40]. The T&CM health services and practitioners do not receive any budgetary support from the national government but rather fund their activities by themselves.

### Study population, sampling and recruitment of respondents

This study included THPs practicing in the eight Acoli districts in northern Uganda. Both men and women aged 18 years and above were eligible to participate in the study. Purposive sampling was used to recruit respondents. A list of THPs registered with Ker Kwaro Acoli as herbalists and or herbalists and diviners was obtained from the secretary of the Prime Minister

of the cultural institution in Gulu. Snowball sampling was also used, whereby the enrolled THPs were requested to identify other THPs to participate in the study. Snowball sampling approach helps to identify respondents that would otherwise be missed by the researchers [41–43]. In addition, the study team also approached chiefdom Chiefs to suggest and provide contacts of people in their jurisdictions they knew provided traditional health services. We recruited THPs who were mentioned by at least two independent persons; either two THPs or a THP and a chief. The THPs were contacted on their mobile phones; those whose phones were not available were visited by the research teams at their homes. The THPs were told the purpose of the study and were provided the information sheet and consent documents regarding the study in both English and Acoli language. Thereafter, dates and places of interviews were set. The research team confirmed the appointments for interviews about two to three days before the appointment dates. We excluded THPs who reported only practicing rituals, prayers, ostracizing spirits and or sorcery as their main healing approaches.

## Data collection

Data collection was conducted between January and June 2018. On the days set for interviews, the research team provided detailed oral information on the study objectives, inclusion criteria, and consent procedures. Written informed consents to participate in the study, and additional verbal consents to audio-record the interview proceedings were obtained from each prospective respondent before data collection. Two research assistants (male and female) conducted interviews at the homes of the respondents. Interviews were conducted in Acoli language in order to gain access to the issues within their contexts. The interviews lasted between 60 and 120 minutes. The research assistants audio-recorded all the interviews. They also made field notes on nonverbal communications, and other events e.g. coming in by a child/visitor, during the interviews. Data collection was supervised by JA.

A semi-structured interview guide was used to collect data. The study guide was piloted with three THPs. Data from the pilot study were transcribed and analyzed manually to obtain a sense of the appropriateness of the tool. The study tool was then refined on the basis of the pilot data [44]. The main issues explored in this sub study were from thematic areas 1&2 of the tool. Thematic area 1: 1. Kindly share with us how one becomes a THP. How do people obtain healing power and hence become traditional health practitioners? 2. How did you become one? How did you obtain your healing power? 3. How is healing power passed on from one person to another or generation to generation? Thematic area 2: 1. Diagnosis of ill health in Acoli. Please tell us how illnesses were/are recognized, how a particular illness was/is confirmed, and how misfortunes (*kec kom*) are diagnosed. 2. Kindly tell us the different ways in which people contract illnesses or become sick in the context of the Acoli people. 3. How do the people (THPs and ritual leaders) who diagnose illnesses differentiate the causes of illnesses or sicknesses and hence select or recommend appropriate methods of treatment and healing?

## Data analysis

Interview recordings were transcribed verbatim by the same research assistants who conducted the interviews. Transcriptions were done within a week of the interviews when field experiences were still fresh in the minds of the research assistants. The research assistants included relevant information from their field notes (in brackets) to what the respondents were saying. Before coding of data, and as part of quality assurance, ADM listened to two randomly selected interview recordings and reviewed two field notes. ADM then read all the transcripts to immerse himself in the data in preparation for data analysis. ADM developed analysis codes based on four transcripts. COG also read two transcripts and independently

developed analysis codes. The final set of codes was agreed upon through consensus between ADM and COG. Transcribed data were exported to ATLAS.ti version 9.2 to support analyses. Iterative thematic analysis approach was used in data analysis [35, 45]. Analysis codes were applied to the corresponding meaning segments as each transcript was read line-by-line. The aggregated coded meaning segments were retrieved from the software, and manually sorted to obtain themes and subthemes. Typical quotes were extracted to validate the respective subthemes. The quotes in the results section have been accompanied by respondents' number, sex and age group. Additional voices to the identified subthemes are included as text boxes.

## Results

### Respondents' characteristics

The median age of the 22 respondents was 59 years (range: 39–80 years). Most respondents were male, and all respondents were married. Majority had attained primary level education (Table 1).

**Table 1. Respondents' characteristics.**

| Characteristics | Respondent's Sex | | Total |
|---|---|---|---|
| | Male | Female | 22 |
| **Age group (Years)** | | | |
| 30–39 | | R1 | 01 |
| 40–49 | R2, R4 | R3 | 03 |
| 50–59 | R5, R6, R7, R8, R9, R10, R11 | | 07 |
| 60–69 | R12, R13, R14, R16, R17 | R15 | 06 |
| 70–79 | R19, R20 | R18 | 03 |
| $\geq$ 80 | R21 | | 01 |
| Missing | R22 | | 01 |
| Median age: 59 (39–80) | | | |
| **Marital status** | | | |
| Married | | | 22 |
| **Educational attainment** | | | |
| No formal education | R5 | R18, R22 | 03 |
| Primary education | R6, R9, R16, R19, R20, R21 | R1, R15 | 08 |
| Secondary education | R2, R7, R8, R12 | | 04 |
| Advanced level education | R11 | | 01 |
| Tertiary/university | R14 | R3 | 02 |
| Vocational training | R4, R13, R17 | | 03 |
| Missing | R10 | | 01 |
| **District of residence** | | | |
| Agago | R8, R16 | R18 | 03 |
| Amuru | R10, R20 | R22 | 03 |
| Gulu | R4, R9 | R3 | 03 |
| Kitgum | R5, R12 | R15 | 03 |
| Lamwo | R6, R19, R21 | | 03 |
| Omoro | R17 | R1 | 02 |
| Nwoya | R7, R13 | | 02 |
| Pader | R2, R11, R14 | | 03 |

R–Refers to respondents' numbers

### Themes and subthemes

The themes that emerged from analyses included; (i) how people gain healing powers and become traditional health practitioners, (ii) perceived causes of illnesses, and (iii) approaches to diagnoses of illnesses.

### Becoming a traditional health practitioner (THP)

Majority of THPs reported that people become healers through successions from ancestors, visions in dreams, learning healing practices from other healers, and through spirit possessions after falling sick with certain illnesses. Healing powers and recognition of potent medicines were obtained in similar ways (Boxes 1 and 2).

---

#### Box 1. Becoming a traditional healer

#### 1. Inheritance from parents

"Mine dates from when I was young and my father used to send me to get for him medicine or at times we would go together. But I only started practicing and taking it seriously when after my father died, because people started coming home to seek healing, but my father had showed me most of his medicine. There are some that he showed my sister who follows me and he didn't show me", *(R8_Male_50-59years)*.

"My father showed me the medicine for impotence. . . I was just chosen. They just chose me but I also happened to have some knowledge of traditional medicine. I will appoint one child to take over from me, most likely a child who used to help me during harvesting of the herbs and is conversant using the medicine that I give people", *(R20_Male_70-79years)*.

"My father was very six and he passed on. . . So when my father was getting weak, he told me that 'son, my days have greatly reduced, but use this medicine when I am gone' and indeed I am using it. For example if I treat someone and they see that my medicine is effective, they come to me and we have a chat, and if I see that they are trust worthy and are able to help many other people, they I give it to them", *(R16_Male_60-69years)*.

"It is really not the same and it varies from person to person and area to area. So many people have learnt by seeing others practice, parents send their children or they go along with them to gather medicine, so the ability to practice medicine is passed down the generations from parents to their children", *(R10_Male_50-59years)*.

"In our family, we give traditional medicines; I am not alone. . . One starts to give traditional medicine after being close to his father who has been also a traditional healer; grandfather, mother and also any relative; this is how one becomes a traditional healer, so mine is like that. I started practicing traditional healing when I was just 15 years old because my father used to give traditional medicine and also my uncles do give traditional medicine so I took up to theirs. It is the reason why it is called traditional, *'Yat Kwaro'*; it is taken from the elders and the elders give to the younger children so it goes on like that", *(R9_Male_50-59years)*.

"How I acquired this medicine, it was passed down to me by my great grandparents, when they died then I inherited it. My grandfather was very fond on me so there came a

---

time when he was so old and he decided to show me the drugs that he used for treatment. Medicine in Acholi needs the owner, the owner could be old and thinks he won't last a long time, then he/she can call one of his children or somebody that he likes and shows him his herbs. Most cases they show it to someone younger to provide continuity, so that the medicine does not disappear even when the owner dies. However even when he is not going to die he can still do so but it is done officially. That is how mine was transferred to me, I didn't steal from someone else and that is why it is effective and it still works even after very many years of using it", *(R5_Male_50-59years)*.

"I became a traditional health practitioner because of my mother, when she was old and about to die, when started telling me the different type of herbs and the illnesses that they treat. She always took me to the forest and showed me the different herbs that she and how to prepare them", *(R21_Female_≥80years)*.

"But as far as traditional medicine is concerned, my mother was traditional healer and she used to treat very many types of illnesses, so in the process all of us learnt how to give medicine. We were seven; all of us had learnt the different kinds of medicine from her. She used to send us to gather medicine from the bush when were a bit old, by around sixteen years. Most people who give traditional medicine (*lumi yat*) learn from other people in one way or another, but unlike for me and my siblings who learnt from our own mother, the rest have to pay to get medicine. In the Acholi context when someone is interested in having your medicine, in most cases they pay some money, it wasn't really expensive. But the most important of all is the hen and a cock which you give the person who is giving it to you, for appreciating them", *(R17_Male_60-69years)*.

"For a long time, I have been working on herbal medicine because my great grandparents from both sides were medicine men and I took an advantage to try and make investigations in some herbs which they knew and out of those we found that there are some plants which are very effective and good in healing. My parents introduced me to these things when I was still very young, sometimes they could send me to go and collect herbs and my dad was a veteran, a second world war veteran and during their training they were trained how to identify plants which can be eaten which can be used in case of emergency so when he came back from the war in 1945 he started also training me in such things. And some knowledge also I got from my grandparents on both sides and even my mum up to now she is 96 years old and she still shows me some drugs", *(R14_Male_60-69years)*.

## 2. Transfer of healing power from other healers

"My father showed me most of the medicines I now use but not all. When I decided to practice traditional medicine seriously, I decided to learn others too. So I decided to ask other healers to show me the medicines that they knew. So I went to Kitgum Matidi, Orom and even Namukora, to the healers there. That is how I learnt how to treat snake bites, scorpion bites and yellow fever", *(R8_Male_50-59years)*.

## 3. Visions in dreams

"I should say I treat mainly mental illnesses and that was the very first medicine that was given to me in the vision I had in my dream. And that is all I knew and practiced for over ten years, from 1972 till around 1993. I was treating only '*two wic*'.However I got

another vision which showed me medicine for '*yat anyona' (stepping on poison)*. Then I learnt the others from my father because he was still alive then, and I also learnt others from my friends. So that is how I started treating only two wic, and it is what most people know me for", *(R10_Male_50-59years)*.

"I started dreaming about medicine and whenever I dreamt I would not sleep, I would go in the night and harvest it and indeed I would go and find that it is the right herb. I fell sick for a very long time, then there was a man when I was about twenty years, he took me back home. You know back at home they like *baro gweno* (the ritual of cutting open a hen and using its intestines to predict the future). When they did so they told me that I had "spirits of our clan". Then I asked them what I should do. When I was asleep the spirits came and told me that you, let's go and get this medicine, so they showed me medicine in the dream, when I woke up in the morning I went and harvested the same herb that I saw in the dream I took it and I conceived! I just dream about my medicine. One day when I was sleeping I had another dream and the spirit showed me medicine that stops vomiting so I stopped vomiting when I took it", *(R15_Female_60-69years)*.

"When I interacted with one traditional practitioner, he also told me that he also dreams about medicine. For example someone can come to you in your dream and tell you that here is medicine, mix this and this and give it to the sick person", *(R6_Male_50-59years)*.

"I started by becoming very ill, they tried all the means of healing including the modern medicine, then eventually I had a dream. In the dream they showed me many types of herbs which surprisingly I didn't forget when I woke up in the morning, then I went in the bush and harvested. I prepared it as per the instruction in the dream then I got healed. Mine did not require any rituals because it simply came through a dream with clear instructions how to administer the medicine. Rituals are performed only when a traditional practitioner is passing his/her medicine to someone else; that is when they give the owner of the medicine a hen. The ritual varies from area to area, in some place it is a goat while others it is a hen", *(R19_Male_70-79years)*.

"I started work as a traditional health practitioner in 1996 from Kiryandongo when I fell sick the disease made me mentally disturbed after distorting my head it would speak to me. Then they took me to a traditional healer (witchdoctor) who worked on me and I was healed; I started understanding. But this spirit kept on talking to me even when I was no longer sick. Then one day through a dream, the spirit showed me someone who was sick, he was mentally disturbed and told me that for this person to be healed I have to administer herbs to him. So I went and gave him medicine and indeed he is okay; up to now he understands and he is married with children. So from that time I continued giving medicine to people whenever they come to me", *(R1_Female_30-39years)*.

"I heard of some traditional healers who have seen medicine in their dreams. They say that for example when someone is ill and people don't have the medicine, they could have dreams with instructions of how to get the medicine and how to prepare it for the person", *(R17_Male_60-69years)*.

"The knowledge of traditional medicine also manifests through dreams, someone wakes up in the morning and says I dreamt about this herbs God told me to go to this tree and get its roots and use it for treating such and such a disease. Then they go and get the

medicine it can be trees, ant hills and many others and come back with it, that also happens", *(R11_Male_50-59years)*.

## 4. Spirits possessions

"For most witchdoctors the spirit manifests itself as a disease, it is not until the ritual (*Wero Jok*) is done then the spirit starts using them to heal people", *(R17_Male_60-69years)*.

"As for the witchdoctors and *lutak-dano*, this being spiritual does not just fall on anyone. In most cases at least someone in their family must have been one in the past then, with manifestation of illness it possess another person who then have carrying out rituals starts to treat people. They then start consulting spirits and also carrying out rituals", *(R2_Male_40-49years)*.

## Box 2. Acquiring knowledge of potent medicines through dreams and visions

"When my grandmother died I started falling sick regularly and they started performing rituals to heal me, however that was not completed. Then I started having dreams, I started dreaming about herbs so from that time till now I still get those dreams. In the dream, I dream that one of my children is showing me the medicine which I get and give to my patients. . .. That is what happens in my case. I don't forget, I go exactly where it shows me in the dream that is where I will get the medicine, for example if I dream that it is under a pawpaw tree, that is where I will find it", *(R22_Female_MissingAge)*.

"Yes people dream about medicines; it is revealed to them there and then in the dream and when they get it, it will cure the illness for which they dreamt about. Although there are different tribes among the Acoli people mainly traditional medicine is got through dreams. And when they use it indeed works. You know god works in different ways. To the Acoli people, all things come from god; and it is one way of sustaining his people by ensuring that they are healthy, hence the dreams", *(R20_Male_70-79years)*.

"I should say I treat mainly mental illnesses and that was the very first medicine that was given to me in the vision I had in my dream. And that is all I knew and practiced for over ten years, from 1972 till around 1993. I was treating only *two wic*. However, I got another vision which showed me medicine for *yat anyona*. Then I learnt the others from my father because he was still alive then, and I also learnt others from my friends. So that is how I started treating only *two wic*, and it is what most people know me for", *(R10_Male_50-59years)*.

"You dream about a disease and also the medicine is shown to you there and then for the sickness, then you also get healed", *(R9_Male_50-59years)*.

"When I was asleep, the spirits came and told me that you, let's go and get this medicine. So they showed me medicine in the dream; when I woke up in the morning I went and harvested the same herb that I saw in the dream. I took it and I conceived. So my son I

just dream about my medicine. . . it is not as if someone showed me any of them", *(R15_Female_60-69years)*.

"God would talk to people. He would tell them that go to this tree and get part of the tree, it will treat such a disease. At times it would be through dreams that go and harvest from that tree and indeed if you did as per the instructions it would work", *(R12_Male_60-69years)*.

"You know I used to just dream about traditional medicine. When I interacted with one traditional practitioner, he also told me that he also dreams about medicine. For example someone can come to you in your dream that here is medicine mix this and this and give it to the sick person. When someone is sick and indeed when you give it to them they will heal, then you continued giving the medicine to whoever is sick", *(R6_Male_50-59years)*.

"Even my mother just dreamt the medicine that she had and then she passed them to me; most of the people dream", *(R21_Male_≥80years)*.

"It happens; and a majority of the medicine we are using today was obtained through dreams in one way or another. So when your father is old and about to die then he shows to his children. That is how it is transferred, but the original custodians of the medicine dreamt about", *(R7_Male_50-59years)*.

"Yes my father dreamt about most of his medicines and that was one way how traditional healers obtained medicines; most of the traditional practitioners of the past just dreamt. It is not easy to understand indeed but when the spirits want you to advance the cause of helping the community, they will indeed communicate to you in dreams. And that is how they communicated to people in the past", *(R8_Male_50-59years)*.

*"To practice traditional medicine one acquires powers in various ways; for some they simply dream or have visions of the different types of medicine. The traditional Acoli acquired medicine in that way, and then that person when he/she becomes old, she/he then passes it on to her/his child/children. And the children too when they become old they pass it to their children or grandchildren. For others, it started as a disease and then they say go to such a place and you will get healing, and at times such instructions also come through dreams, then when they get the medicine, they tell people I got medicine that can treat such and such a condition. That is how people acquire the power to practice traditional medicine. One does not just wake up one morning and start practicing traditional medicine or starts giving herbs, but it comes through a dream and eventually spreads to their descendants", (R19_Male_70-79years).*

**Successions from parents and grandparents.**   Majority of respondents reported that most people who become healers inherit healing powers and traditional medicines for specific illnesses from their parents or grandparents. When one inherits healing power and medicines for treating particular illnesses from their parents, no ritual for transfer of healing power is performed. The parents/grandparents simply show the medicines to the children or grandchildren who had often worked with them. This handover is often done when the parents/grandparents are of age, or sickly and likely to pass on soon.

*"People obtain healing powers if their grandparents also practiced traditional medicine. In my own case my paternal grandfather was a traditional healer, my father was also a traditional healer. When he was about to die he called us his children and told us that when I die, this medicine should help you in future. He told us that this medicine if someone has been producing only girls, you get it and pound it; but it is boys who are supposed to pound. Then you give it to the woman who has been producing only girls to take"*, *(R18_Female_70-79years)*.

*"For me I got my medicine from my father. When your father is about to die, then he shows to his children; he comes and spits on your palms, then you clean the spittle on your head, then he tells you that 'you can use this medicine, I don't have energy any more'. Then you start using it when someone with that particular illness comes to you. That is how it is transferred, but the original custodians of the medicine dreamt about. So that is how I began practicing this medicine. That was a long time ago, in 1960"*, *(R7_Male_50-59years)*.

*"Most renowned traditional healers, the first thing about them is that either their mother or father was a traditional healer. So they keep on learning as they go along and at times the parents don't intentionally pass on this knowledge. But as they carry out their daily responsibilities of collecting herbs they go along with their children or send them to collect them in the bush, then keep on learning"*, *(R11_Male_50-59years)*.

**Transfer of healing power after illness episodes.** When one falls sick or gets afflicted by an illness that is unusual and difficult to treat, he/she can request the healer who worked on him/her to show him/her the medicines and how it is used so he/she can also treat others. When the healer accepts this request, then the healer-to-be makes some payments in kind and a ritual for transferring healing power is conducted. The medicine fails to work if one starts using any medicine for treating others before the ritual of transfer of healing power is conducted. The established healers often know the nature of the strange or unusual illnesses that foretell of a healer-to-be; sometimes the healer who manages to get the cure for the illness experiences a strange dream that tells him/her of the healer-in-the-making who is sick and who requires a particular treatment regime. When the senior healer follows the dictate of the dream, administers the revealed medicines, and the person gets healed, then the healer knows that the sick person is destined to become a healer, and so the transfer of healing power rituals are conducted to ordain the sick person a healer.

*"I started practicing by giving people medicine to treat snake bites. I was not the first to use it, but a snake had bitten me then I called someone with the medicine, then he treated me and I was healed. Then he told me that if I wanted to have the medicine, he could show me. But I should give him something in return. When I told my parents, both of them were still alive by then. They decided that his medicine works, so my father gave a goat to the owner of the medicine and he showed me the medicine and how I should use it, and I am still using it to date"*, *(R16_Male_60-69years)*.

It was a social and community responsibility to pass on medicine for healing from one person to another. It was an obligation on the healers to ensure that they do not die with the healing power of their families/clans; they had to pass it on so the society continues to benefit.

*"If you want people to get healing in the community then you have to transfer it to someone else. In Acoli when someone wants to have your medicine normally you ask for a hen, others ask for a goat; so if you give them then they can show you the medicine. That is how the Acoli pass on things like medicine"*, *(R12_Male_60-69years)*.

The person receiving the healing power and who will be shown traditional medicines that are potent will bring a chicken or a goat of specified colors, usually white and without spots to signify purity of heart on the part of the giver, and that the purpose of the power and medicines is clean, and never to be used for harming people.

*"Just like I learnt how to give medicine from my Auntie, there are people who also learn from others. I learnt most of mine while assisting my auntie, however when I left her home and I wanted to start giving people medicine officially, I went back to her and told her so. Since mine is not spiritual, she told me that she would purify my practice (buku yat), so I took to her a hen, a white bed sheet (cuka matar) and money worth twenty thousand shillings. She then gathered all the herbs that she wanted to hand over to me on the white bed sheet and moved the hen round it three times, then the hen was freed into her compound. But as my own initiative I gave a goat which she slaughtered and we ate together with other people",* **(R4_Male_40-49years)**.

*"Medicine was not transferred without performing a ritual. The owner of the medicine would go and harvest the medicine then he hands it over to the person he intends to give it to. Then the person receiving it hands him a cock which is then used to circle him three times for a man and four times for a woman. Then the owner of the medicine says use it to help people, I have given it to you with a clean heart",* **(R5_Male_50-59years)**.

**Visions and dreams.**   People without tradition of healing in their families can get healing power and become THPs through visions and dreams. They experience particular dreams that tell them about how to heal, either their own illnesses or of someone else. They visualize the appropriate medicines and how to use them in the dreams. They then use the medicines, and if the patient gets better, they then become experts in the treatment of that illness.

*"There are people who dream about medicine; for example my father would dream about it, that is why he was able to treat very many illnesses. He was also able to give me his medicines when he was very old and he thought he would not live for long. For example he showed me medicine for 'two Rubanga' (TB of the backbones) and not many traditional healers have that in Acoli; just a couple of us are able to treat people with 'two Rubanga'",* **(R8_Male_50-59years)**.

*"My paternal grandmother was not a witchdoctor. But she had a spirit called 'Jok Tak' and she also had a tortoise which has now transferred to my place; it lives under my bed. So my grandmother used to go and get herbs from the bush and give people. . . and many people got healed. When my grandmother died I started falling sick regularly and they started performing rituals to heal me, however that was not completed. Then I started having dreams, I started dreaming about herbs; so from that time till now I still get those dreams. In the dream, I dream that one of my children is showing me the medicine which I get and give to my patients",* **(R22_Female_Missing_Age)**.

*"The genuine traditional medicine practitioners that I know dream about medicines. . . spirits lead them in the dreams and show them medicine which they then master and use",* **(R10_Male_50-59years)**.

**Spirits possessions.**   Majority of THPs said that some people gain healing powers during a ritual of dancing and singing to their ancestral deities, '*wero jok*'. This ritual is often presided over by a witchdoctor on a person who has been seriously ill and who manifested evidence of spirits possessions during the illness episode.

*"There are people who started practicing traditional medicine when rituals to chase away spirits have been performed (wero Jok). Such spirits manifested through diseases and they would then take that person to a witchdoctor who prescribe it either as 'Ayweya', 'Kulu' or any other kind of 'Jok'. They would then organize for a ritual to be performed, after which the person gets powers to practice traditional healing. Sometimes the ritual could take days",* **(R8_Male_50-59years)**.

*"It started when I was still a young boy. I was possessed by a spirit which would throw me down. For example, I would be playing with friends then all of a sudden I would fall down and become unconscious; this happened till I was about twelve or fourteen years. My mother and father moved to different witch doctors enquiring if they could heal me... They consulted widely with our other relatives and then they bought a sheep, a white cock and other items. They took them to one particular witchdoctor who carried out the ritual (wero jok). So after the ceremony was done, I felt as if a heavy luggage was placed on my shoulder, when I told that to my parents and they told the witchdoctor, he told them that it means I should start practicing traditional medicine because even the spirits had manifested in his Gagi (cowry shells). He then gave me a spear, a drum and ostrich feathers (yec wudu). So that is how I began practicing. But when I became older, that is when I learnt more medicines, a couple of them through dreams and others I learnt from other people",* **(R2_Male_40-49years)**.

## Perceived causes of illnesses

The majority of respondents reported that they treat illnesses and patients based on the perceived causes; that their medicines target the causes of the illnesses, and the success of their work depends mainly on the proper identification of the causes of illnesses. They said illnesses are caused by several factors interacting with each other, including individual fate, spiritual forces, bewitchment, contagions, transgressions of cultural norms and curses.

**Fate and natural causes.**   Majority of respondents associated illnesses and diseases to fate; that people who fell sick were meant to do so by design; that was their fate, not because they did something wrong or did not do something protective.

*"What causes illness is normally God's will; you know it is God who decides all things in the world. Even diseases, it is God who causes it. There is no one who is responsible for causing diseases. Except for illnesses like HIV and Syphilis which I think man looked for intentionally by sleeping with animals",* **(R20_Male_70-79years)**.

*"Most diseases occur naturally. Someone would get sick just because they were exposed to whatever causes the disease. But because all the time we think something cannot happen without a reason, we tend to suspect other people and at times it may not be true. People can harm others, but not all the time",* **(R2_Male_40-49years)**.

**Spirits attacks.**   Illnesses and diseases were also thought to result from the invasion of a person or family by disgruntled spirits of the dead. People are to be cared for during illnesses and bad times by their families. If a deceased person felt they were not cared for well enough by the family members, and they died while in that kind of disposition, their spirits sometimes haunted the living. Occasionally, the living members of such a family experience certain illnesses including mental disorders that are directly attributed to the actions of the disgruntled spirits. The voices of respondents are below and in Box 3.

Box 3. Perceived causes of illnesses

1. Spirits attacks

"There are also spirits of dead relatives who for example if they die away from home or in the wilderness and their bodies were not brought home or their spirits were not appeased. These spirits would cause sicknesses; they would attack a family member, it would keep on attacking people till the spirits are brought home", *(R6_Male_50-59years)*.

"Spirits and "*Cen*" (angered spirits of the dead). They could attack someone and cause serious illnesses and it would need rituals. It would require a goat to appease the spirit. In the past they used to call a witchdoctor; the witchdoctor would determine what was required for the ritual", *(R21_Male_≥80years)*.

"Also killing people anyhow; the spirit of the dead person will keep on haunting the family that caused his/her death causing illness and death. The Acoli of the past would look for such acts and solve it. For example if we are brothers and I caused your death or the death of your child and rituals aren't done to its logical conclusion, people will keep on dying in that family till a ritual is done. It is still happening even today", *(R13_Male_60-69years)*.

"I know most young people don't believe this; but it is a very big mistake not to take this seriously because spirits exist and they cause disease. Don't get me wrong, there are both the bad and goods spirits. Culturally we go through the good spirits to fight off the bad ones, and that is how our society has been able to survive for long", *(R2_Male_40-49years)*.

2. Curses

"Curses also cause diseases but only when you have wronged someone. In our culture, it is only specific people who can curse you, for example your paternal uncles. They can curse you that; you will never get a child in your entire life, or since you have beaten me, you will continuously keep on fighting", *(R22_Female_MisingAge)*.

"If you are cursed, you will get ill and die if measures aren't put in place to resolve it immediately. Curses are very bad, it can cause death", *(R20_Male_70-79years)*.

"Some diseases were caused by curses, your stomach can swell when cursed and not only that, we talked about misfortune earlier, it is also partly caused by curses, when you offend someone they can curse you", *(R6_Male_50-59years)*.

3. Witchcraft

"It is normally transferred by witchdoctors and some people use them as a means to torment their enemies or others that they don't like for reasons best known to them. They send spirits to attack and cause disease to others", *(R22_Female_MissingAge)*.

"There were also illnesses that were sent to someone by their enemy or people who are jealous or bad hearted. You see in Acoli there were bad as well as good spirits, and a witchdoctor could sometimes have them both. So people utilized them to send diseases

to other people, for example some forms of headache, swollen stomachs and other diseases too", *(R8_Male_50-59years)*.

"Witchcraft is there, if someone sees that you are really a hard working person and you also know that you work hard, then you get much money or even have very much property, you have enough things to eat so the witch won't accept to see that; if they see they will just get annoyed instead of being happy, then such can come and bewitch your eyes, will bewitch any part of your body", *(R9_Male_50-59years)*.

"Many people also get bewitched, for example a very bright student can get mental illness all of a sudden, and that is being bewitched. They send spirits that affect the head (Nweno wiye)", *(R17_Male_60-69years)*.

### 4. Malevolent actions

"For diseases like "*Two Rubanga*" people just step on them "*Ki nyono anyona*". When someone has it, the process of healing it involves bathing the patient along the road, so if someone else steps on the bathing spot they contract "*Two Rubanga*", *(R22_Female_MissingAge)*.

"I am forgetting the other one that people can contract, "*Abar wic lela*" or tension headache. That one can also be passed from one person to another. If you step on a razor blade used to treat a person with that condition then you will contract it, because they throw the razor blade together with some money along a path which is rarely using, if accidentally someone finds it then they will contract the disease", *(R22_Female_MissingAge)*.

"Two Agoobi" which causes bleeding in women; people transfer it intentionally. A woman who has it intentionally touches the waist of another woman and she will also contract it. They just torch your waist in a joking way", *(R22_Female_MissingAge)*.

"We also have diseases that are sent like cataract, *kelle*; this affects the eyes. But you should also note that there are some forms of cataract that are not sent, those ones can be cured medically, but the one that are sent cannot be healed medically, they keep on reoccurring. I know people who can effectively heal it", *(R10_Male_50-59years)*.

### 5. Poor hygiene

"I am forgetting causes of disease like poor hygiene, it is still even common to find people suffering diarrhea just because they have been unable to keep their homes clean", *(R2_Male_40-49years)*.

### 6. Poisons

"Conflicts in the community also led to disease. . . people gave poison to their enemies. . . People even feared eating and drinking together for fear of being poisoned", *(R13_Male_60-69years)*.

### 7. Poor diets

"Poor nutrition also caused illness in children that is why many of them would have swollen stomachs and brown hair", *(R2_Male_40-49years).*

"In my opinion people in the past ate very nutritious food; they would not fry food, and most food was pasted; milk was plentiful. But at the moment people fry food and eat processed food which is one of the causes of diseases. During the time of our parents, people were very strong, even alcohol, we had only local brew, we didn't have alcohol in sachets like today", *(R16_Male_60-69years).*

*"In the past when someone died and the spirit was not appeased, it would cause diseases to its relatives. It was a mechanism of ensuring all the rituals were done in accordance with the culture. They would convene as a clan in the presence of a witchdoctor and elders, they would then tell all the clan members that, the spirit of so and so was not appeased. And they would summon the spirit to talk to the people (Laughter). [. . .] They would call the spirit home",* **(R16_Male_60-69yars).**

*"Spirits indeed cause diseases. . . Such spirits can attack children or even adults in that home. They are spirits of the dead within the family who were not happy with the way they were treated prior to their death. It can also be spirits of people who don't belong to that family but met their death because of someone in that family. The witchdoctor can capture the spirit in a calabash (opoko) and burns it and then it ceases to cause problems to people in the family",* **(R20_Male_70-79years).**

The spirits attacks sometimes occurred following inadvertent stepping on the remains of dead people or just seeing such remains in deserted places. When young people accidentally come across remains of a dead person, they were expected to report to the elders immediately and have a cleansing ritual conducted so that the spirits of the dead person/people they encountered do not torment them.

*"Mental illnesses for example, people who find a dead body in the bush ought to tell the elders so they are cleansed; if not they will be attacked by the spirits. Even ancestors who are not accorded a decent burial can cause illnesses and even death. Even when you kill somebody intentionally or unintentionally the spirit will attack you, unless rituals are done and compensations (culu kwo) paid. . . In some instances if the other family is known, culu kwo is even encouraged and should be conducted immediately",* **(R10_Male_50-59years).**

*"To the Acoli, most illnesses occur because of a reason. Mental illness is always associated with bad spirits (gemu) which can only be healed by performing rituals. So if a person came into contact with those spirits, either by moving at night, seeing a person who could have committed suicide, or even by killing a person, they can develop mental illness",* **(R4_Male_40-49years).**

A substantial minority of participants reported that there were bad spirits that would attack several people at once, affecting the whole community, causing outbreaks of diseases (epidemics) including measles and meningitis.

*"Culturally we had what we called "Jwee"; these are spirits that would attack an area at a particular time. An example is measles or meningitis, or cholera. . . People would tell each other 'Jwee has come, let us live in harmony, and let us not quarrel among ourselves'. It was so bad*

*and if it came into a home, if not controlled it could kill many people", (R13_Male_60-69years).*

**Curses.**   People were expected to conduct themselves in some culturally prescribed ways. If someone deviated from these cultural norms or conducted him/herself in manners that angered other people, especially the elders, they could be cursed. The curse would follow them and cause them some ill health or misfortune.

*"Curses cause illnesses. . . my maternal uncles can curse me! They could curse me and I urinate in the house even when I used not to; they can curse me and my manhood fails to function even when it was perfectly alright. . . They can forgive, however one would have to take a he-goat and other requirements to undo the curse", (R16_Male_60-69years).*

**Witchcrafts.**   Witches and sorcerers are feared for causing strange illnesses. They are considered to have some powers that they can send onto people, leading them to fall sick or develop some disorders.

*"Some witchdoctors do, the real ones do, and they can send illnesses as well as heal. Or someone else can use a witchdoctor to cause illness onto another", (R16_Male_60-69years).*

*"Acts of witchcraft (tyet) also caused diseases and this was used mainly in inflicting harm on ones enemies. However, a jealous person could also use it just to harm someone", (R4_Male_40-49years).*

A respondent disagreed that witchcraft causes illnesses. He contended that people would simply be in denial.

*"Let me tell you, there is no connection between sorcery and disease; for someone to cause harm maybe they have to use poison which will then cause problems in your body. But I can't send something in the air to harm anybody; that is not practical. Most people say that they used sorcery on them. That is not true, sorcery doesn't work. Those people probably have some complications in their systems and in most cases people with HIV first complain that they have been bewitched", (R11_Male_50-59years).*

**Contagion and infections.**   Some respondents attributed causes of ill health to the modern biomedical theories of contact and infections or infestations by some pathogens.

*"There were very many ways of contracting illnesses and almost every disease had a different way of being contracted or spread. Guinea worms (two coo) were contracted in water points like wells and streams. When people went to fetch water or took animals to drink water in an area infested with it, then the chances were high that they could contract it. Some diseases were also contagious for example measles, if someone had it, the likelihood of others getting it was also high", (R8_Male_50-59years).*

*"Some illnesses are also contracted through contact or by staying close to the person who has that same disease; for example, eye infections (lit wang) or even measles among children", (R4_Male_40-49years).*

*"Yes some diseases come through the air, for example Measles "anyoo"; it comes by the wind. For example, if you went to a home were someone had measles, chances were high you would contract it. That is why in the past if someone had measles in the home, they would mount*

*two poles and a cross bar over them right at the entry into the homestead to warn people not to enter in such a home; reason being they would transfer it to wherever they would go next", (R19_Male_70-79years).*

Sexually transmitted infections were recognized by majority of respondents as a common illness among the Acoli.

*"Having multiple relationships can also make one contract sexually transmitted diseases that can make a woman infertile", (R15_Female_60-69years).*

*"We also had diseases that were contracted through sexual intercourse; gonorrhea and syphilis are not new, they have always been around, and some people even think that one of them could have evolved into HIV. So gonorrhea and syphilis was contracted mainly by the youth but not limited to them; even elders who didn't respect themselves contracted it", (R13_Male_60-69years).*

**Poor hygiene.** Poor hygiene was considered a common cause of illnesses that often get transferred from one person to another.

*"There are different diseases that people can contract, and most of them are due to poor hygiene. Women should not wear wet clothes; if you put it on, it is possible to contract infections through the female organ and from there it attacks the kidney [. . .]. Wearing wet underwear, especially men who are not circumcised; you find some whitish dirt in the foreskin (adwooka) and when they have unprotected sex it goes together with semen into the uterus, it causes infection and rashes in the women.", (R15_Female_60-69years).*

*"There are diseases that people contract because they don't clean their surroundings, they eat unclean food, and they don't clean their finger nails so it keeps dirt. . . They don't clean areas around the home, so the surrounding becomes bushy and harbor disease causing organisms", (R11_Male_50-59years).*

**Heredity or genetics.** A few respondents reported that some particular illnesses and disorders occur in particular families and clans. They attributed these illnesses to heredity or genetics.

*"Other diseases also run in the families in certain clans (kaka). So when a person who has such a disease dies, the disease will remain and continue attacking other people in the clan. We also carry out rituals to cleanse such a clan, we talk to the clan members that this disease be chased away from your clan", (R8_Male_50-59years).*

Some of these illnesses that are thought to run in families shape the manner people marry. Elders would caution their sons and daughters never to marry or get married into those families with such illnesses. The family gets stigmatized and discriminated against. In a way this was to deny such people reproduction and opportunities to pass on the illnesses, and perhaps bring an end to such illnesses.

*"Other diseases are hereditary. It keeps on passing from one family member to another. For example, there are families that have had issues of mental illnesses over a very long period of time, to the extent that people are warned not to marry from such families" (R10_Male_50-59years).*

*"It is not easy to prove but there were families that were sickly. . . through rumors people did not want to intermarry with them [. . .]. Similarly, there were families and clans whose grandparents and elders could have murdered people in the past and the spirits of the dead person/ people caused illnesses to them and this always stayed in the family. Some curses (Lam) are also generational; they are passed on from generation to generation. For example, we have a family here whose children don't stay in marriage, they keep on getting divorced and the root cause is that their great grandfather beat and killed his first wife, so the wife's family cursed him—that his offspring's will always be like him", (R4_Male_40-49years).*

**Poisons.**　Poisons cause various illnesses especially those associated with vomiting and abdominal distension. Poisons would intentionally be put into drinks and foods to harm someone or some people. Occasionally, the poisons are drunk or eaten accidentally.

*"There were also people who would poison people in the community but there were also others who had the antidote for such poisonous substances", (R20_Male_70-79years).*

*"People also became sick when they drank/ate poisons; those days it would happen at drinking places, and other gatherings. . . Some people would also step on medicine "nyono yat" or they would touch on it (Mako yat)", (R2_Male_40-49years).*

**Malevolent actions.**　A few respondents said that some illnesses would be contracted through malicious acts of other people. They intentionally pass on the illnesses in some way to other people, just so that the other people also get afflicted.

*"There are also people in the community who we called "Luyir" (bad eye). Such people would just look at you and something bad happens to you, for example your eyes can develop cataract "kelle" or your stomach swells or any other disease; such people were also there. But that was easy to deal with; we would confront the person suspected of doing it and if it is really them, then they would simply say if it is me who has caused this problem to you, be healed. You see just by saying that then the person would be healed", (R6_Male_50-59years).*

**Poor diets.**　Majority of respondents believed that poor diets especially involving foods that are highly processed or not directly from the gardens could cause certain illnesses especially cancers, diabetes and high blood pressure.

*"In my own opinion people have been eating things which they are not supposed to eat. People have left their original food crops. People have resorted to grafted foods which are causing cancer", (R3_Female_40-49years).*

*"The foods that we used to eat like Lakilikili, ocuga, obajara, and many other different kinds of foods are not common now. . . These days people like cooking oil; if you don't use cooking oil, people don't enjoy your food. . . these oils are not good for health. In the past they used to paste all forms of food including chicken. Meat was always boiled just with salt and water. Vegetables were also boiled. Things like malakwang, such foods enabled longevity, and people lived for so long", (R12_Male_60-69years).*

**Abrogation of cultural norms and misdemeanors.**　Whenever people conducted themselves in manners that were contrary to the norms of the clans, certain disorders or illnesses would follow. For instance, certain items and regalia were not to be used for any other purposes, otherwise illnesses would strike someone.

*"That calabash which was used to protect the baby strapped on the mother's back from sun-shine should not be used by a stranger who could have killed someone; if they used it for drink-ing water or anything, the child would fall sick. So Acoli had a lot of cultural practices which if not followed can cause diseases. But if people observed those cultural practices, then there wouldn't be diseases", (R5_Male_50-59years).*

### Diagnosis of illnesses

Most respondents said one of the main challenges in treatment of illnesses is identifying the actual causes of illnesses in the particular patients. They said their medicines are administered based on the root causes of the illnesses (Box 4). Traditional health practitioners do seek for the root causes of illnesses in different ways including consultations with the spirits, observing trends of events and occurrences, analyses of symptoms and signs, and eliciting a positive history of the illness in the family or clan of the patient.

---

Box 4. Diagnosis of illnesses

Diagnosis of illnesses

1. Consultation with Spirits

"There are instances when a person is abnormally quiet and they look depressed and we have to consult the spirits and tell us the cause of the illness", *(R10_Male_50-59years)*.

"You see in the past, some witch doctors were very bright, but these days most of them are after money. Those days they would take someone to the witchdoctor and would consult the spirits which would then tell the cause of that disease, they would then rec-ommend the appropriate treatment for that condition and in some cases they witchdoc-tors also doubled as herbalists and give the appropriate medicine for the condition. But they would also refer to herbalists in case they didn't have the medicine for that condi-tion. Those days we had real witch doctors not fake ones like the ones of these days. That is true and he would consult the spirits to know the exact cause of the disease", *(R12_Male_60-69years)*.

"You see we used to worship our God but we would do it by going to the witch doctor, then he would tell us the cause of all those conditions and how to stop them from caus-ing harm. Witch doctors were a medium of communicating to God", *(R13_Male_60-69years)*.

"It is only witchdoctors who consulted using spirits, who would almost certainly diag-nose a disease. The rest of the healers used to look at the signs and symptoms. I am not saying that it wasn't effective, yes it was and many people were healed", *(R17_Male_60-69years)*.

"In Acoli, the main way of recognizing illnesses is by observing. We observe the signs of the illness, for example coughing, diarrhea, fever and so on. But in cases were the signs are not clear, I can consult the spirit and listen to what they have to say. Oh, (laughter), we have our ways, I use cowry shells, coins and we have other things that we work with. It is easy to differentiate these diseases and not only through the signs and symptoms but

---

the spirits can tell us when we throw the shells (Ka wa onyo gaggi). Also the pattern the shells take when thrown indicate the condition”, *(R2_Male_40-49years)*.

## 2. Symptoms and signs of illnesses

“The major disease that we treated here was Malaria and it would come with very high fevers, and the patient would sweat a lot, and sometimes they would also have diarrhea which is yellow in colour. The person would also be shaking; then we would know that it is Malaria. Yes we would observe and then then we would ask the patient, that do you feel headache, they would say yes and joint pain then we would confirm that it is Malaria they are suffering from”, *(R6_Male_50-59years)*.

“There are symptoms which can obviously be seen, maybe you are sneezing, maybe you are coughing and maybe you have headache; all these show some imbalance in the body. And even the body how the skin looks like; sometimes you are emaciated or vomiting like that, so there are many ways to identify sickness”, *(R14_Male_60-69years)*.

“If you have a snake bite we will definitely know because we shall see the spot of the snake bite. Snakes bites also cause pain, paralysis and can send the person it shock attacks it delays untreated. Also in case of poisoning the mouth and the tongue will turn black, and the person vomits. And in case of impotence when you go to have sex, you will realize when you can’t erect. We know, for example if it is a spiritual attack we differentiate from the signs because since time immemorial our forefathers always described for us the signs of this spirit attacks and it is passed down to the preceding generations so that is how we know. It is easy to tell that a certain illness is due to spirits because spirits cause mental illness, they distort the persons understanding, he/she speaks uncoordinated words and eventually if left untreated it can put the person down. That is the difference with other diseases. So that is how we know”, *(R7_Male_50-59years)*.

“In the past illnesses would be observed/seen. For example Guinea worms started by itching the area around which it is located then eventually it comes out. Malaria would always come with very high fevers and headache. Typhoid also brought a lot of fever. . . Epilepsy caused jerking and the person suffering from it foamed at the mouth. That is how we could differentiate the diseases”, *(R19_Male_70-79years)*.

“We see the signs, for example I told you about chicken Pox. It starts with very high fever and burning of the body and they caused boils and then blisters. When I see that, then I just know. . . all the illnesses come with signs”, *(R17_Male_60-69years)*.

“For *two Lango* (disease attributed to a neighboring clan) you will see wounds in the mouth of the child and also in the anus of the baby. The child will have shown some signs of fever a while earlier”, *(R5_Male_50-59years)*.

“We also have another disease that has been named by the whites; it is called cancer. We know it by the way it appears, the way it manifests and the way in affects people [. . .] Even what you excrete isn’t normal; the urine and feces. Also we see that on women there is bleeding and severe pain that doesn’t stop easily, that is when we know it is cancer. So to enable us differentiate, the signs makes it easier for us”, *(R9_Male_50-59years)*.

## 3. Use of animals, birds and objects to diagnose illness

"I use cowry shells (onyo Gaggi); after that, I get coins. I have coins even the coins which have the portrait of Jomo Kenyatta; I still have them. Now I pick each and spit on each while making incantations; let me see if this disease is there, let me know if this disease is there. I throw cowry shells, if it is not there it will definitely show this other one. Then I will show you which disease you have. I ask that you disease what do you want that you are attacking my son/daughter, then it will also show itself and we determine the medicine to give. Then the patient will be free", *(R15_Female_60-69years)*.

"There are also some people who look at the palm for tell what is wrong with the person. They can tell from the lines on the palms and other things that I don't know. They diagnose things like bad luck/misfortune (kec kom) and most of the illnesses caused by spirits", *(R2_Male_40-49years)*.

### 4. Laboratory testing

"If I find a disease which I can't understand, I first take them to the hospital to carry out some tests to determine the type of disease they are suffering from. I am the one who pays for the tests", *(R15_Female_60-69years)*.

"We sometimes depend on the hospitals to test people to determine what is wrong with them, at times when they do so, we can give them the right medicine", *(R2_Male_40-49years)*.

**Symptoms and signs of illnesses.** Majority of respondents mentioned that they diagnose illnesses based on critical analyses of the signs and symptoms the patients present with. They said particular illnesses would commonly have certain signs.

*"Diseases also come with signs; the challenge is most people don't know them so it is up to the traditional healers to see and tell them... Most signs of illnesses are dizziness, headache, loss of appetite, body weakness and others. So if someone has the above signs then most likely they are unwell", (R4_Male_40-49years).*

*"Many people think we just guess the diseases and then give medicines; no, that is not the case. Because of treating these conditions over and over again, we can tell from their signs. Most diseases that show by high fevers, vomiting and sweeting are Malaria and Yellow fever. So when I see any of those signs, I know what to give the patient. 'Two Rubanga', causes chest pain over a long period of time which can then limit activity of that person. If left untreated, it can permanently disable the person. So in reality that is how we identify diseases, by seeing due to experiences acquired over time, by consulting the spirits", (R8_Male_50-59years).*

**Consultation with spirits.** When the signs and symptoms, or history of illness is not leading the healer to a particular illness or disease, they consult the spirits, either directly through use of some items or through the diviners as medium of communications with the spirit world.

*"We also consult spirits to find out diseases that were/are afflicting people in the community; of course there has to be a reason for diseases to affect a whole community! For example, if there was an outbreak of a strange disease formerly unknown to people in the community, we consulted the spirits through a diviner, and in most cases they told us its cause... Things*

*required in the consultations include a white cock or a certain other colour, some money, and alcohol; and one had to go with at least an elder", (**R8_Male_50-59years**).*

People who suffer from unusual, strange and repetitive illnesses consult the spirits of their ancestors through the diviners.

*"If you find out that all the things that you are supposed to do or to have as a normal human being aren't working out then probably you have misfortunes and therefore you should seek ways of addressing it. If you are unlucky in all aspects—like you cultivate with other people but your yield is poor or you produce children but they die in infancy. The Acoli would go to a witchdoctor to find out the cause of the misfortune. [. . .] All that was handled culturally and at times it would even call for shifting the family from one area to another because the cause of that misfortune could be associated with one settling in that place. That is why some people sought refuge in other areas", (**R20_Male_7-79years**).*

**Observing patterns of occurrences and events.** When things that happen in the life of a person or a community is different from expected, then elders begin to suspect misfortunes as the cause of illnesses and or abnormal occurrences.

*"Misfortune is as serious as any other illness and it can be life threatening, because it can drive people to commit suicide, or be depressed. If all the time calamity befalls a person or a family for example premature death, low yields, poverty, all these indicate misfortunes. . . Misfortune can run in families, and most times if not appeased then generation after generation will have it. Therefore there are things that ought to be made right in order for this to stop", (**R10_Male_50-59years**).*

**Use of bones of animals/birds and other objects to diagnose illnesses.** Majority of respondents said in difficult circumstances, they use certain objects to arrive at diagnoses. There are signs the healers look for in these objects including the way the objects settle down when thrown up. For example, particular kind of shoes are thrown up and the direction the front points tells of certain particular illnesses. Cowry shells are thrown up, and the way the shells distribute themselves as they settle down (patterns formed) tell of certain particular illnesses.

*"In Acoli. . . We have what we check with, like some use water. There are some people (THPs) who can check in water, there are some people who can check using cowry shells (gaggi). . .. So all those practices are there with different people", (**R3_Female_40-49years**).*

"We still have many Ajwakis (diviners); if you go to one they have cowry shells (Gaggi) which they throw around and depending on its direction, the shells will help them to identify the cause of the illness that this person has", (**R4_Male_40-49years**).

The gut signs or tests are done by opening up a goat or chicken to see the diagnosis for an illness that may have persisted in spite of several treatments. The gut sign is also used in circumstances when the patient presents with uncommon symptoms or conduct not typical of any known or common illnesses.

*"We still have the practice of slaughtering a goat or a hen, and looking at its guts (Baro ii dyel or Baro ii gweno). Those who specialize in this practice have the ability to make deductions basing on the appearance of the gut of the goat or hen", (**R4_Male_40-49years**).*

**Laboratory tests.**   A substantial minority of the respondents send patients with unclear symptoms and signs, or illnesses they are not familiar with, to the biomedical facilities to have laboratory tests to determine the causes of the illness before they decide on the treatments to provide.

*"I also encourage the patients to first visit health centers to help in recognizing the disease and its causes. This is because they have better machines at doing so. Unless the signs are obvious then I would start treating right away", (R10_Male_50-59years).*

*"Personally, I would prefer if the patient has gone to the hospital and a diagnosis has been made. This makes the treatment easier; so I rule out other diseases, or I treat other underlying causes. For example if someone has HIV, I would also like to protect myself", (R4_Male_40-49years).*

**Diagnostic rituals.**   In some circumstances, especially when the illness is considered strange and rare, and has not responded to treatment with common medicines, the THPs consider rituals in addition to consultations with the spirits through the diviners. The diagnostic rituals are often performed when one suspects the cause of the illness to be due to spirits attacks or some sinister evil forces.

*"There is 'wero jok' (dancing and singing to the deities) if the illness is deemed to be caused by spirits. The ritual of 'wero Jok' is done so that the spirit that did not want to come out does so and tells us what is causing the illness. Now this is only for diviners", (R4_Male_40-49years).*

Misfortunes were not only diagnosed based on patterns of abnormal, unexpected occurrences and events. There were also rituals that would be performed to diagnose them.

*"Misfortune was manifested through bad luck, for example if you went to hunt and something bad always happened to you, then we would know you have misfortunes. . . Also if one cannot conceive or is unable to impregnate a woman, or on the other hand you could give birth and your children keep on dying. In that circumstance we would do what we call "alama" or swearing because this death could have been caused by something I did or what my in-laws did. Then they would do something called "kwir", and both sides would perform this ritual to purify the couple. For example if you are our in-law, we also call your relatives as well as our people, then we mix the "kwir" in water and both sides get a hen. If your hen dies as a result of taking the "kwir" and ours survive, then we know that the cause of the misfortune is as a result of something that you or your relatives did. That is one way how the Acoli diagnose misfortune", (R19_Male_70-79years).*

## Discussions

This study has shaded light onto the Acoli traditional health practices, especially as regards how one becomes a THP, perceived causes of illnesses, and how the THPs arrive at root causes (diagnoses) of illnesses. Our findings provides the post-conflict status of THPs practices among the Central Luo of northern Uganda. The authors discuss these findings with respect to results from other societies, but not necessarily in comparisons with THP practices before, during and after the prolonged violent conflict that afflicted the study region.

### Becoming a traditional health practitioner

The majority of respondents revealed that most people become healers through successions from their grandparents or parents. In the Luo tradition, as it is for most African traditional

cultures, the grandchildren or children learn skills through apprenticeship. The grandchildren or children who eventually inherit the healing powers and become healers are those who would have been working very closely with the experienced healers. They go with the healers to the bushes to harvest the herbs; they observe and participate in the preparation of the plant components into herbs; they also observe and participate in the administration of the medicines alongside the healers. By the time the healer grows old and frail, which is often the time they pass on their healing powers, the imminent heir is already well versed with the healing practices and rules regarding healing. The ancestral spirits confirm to the experienced healer whether or not the incoming healer is the right person to inherit the healing power for the clan. Our findings are similar to what Gessler et al (1995) described among the Luo in Tanzania where majority of healers were men, and became healers through inheritance of healing power from parents and grandparents, as well as following episodes of illnesses during which ancestral spirits direct them to become healers, and through dreams and visions [46]. Majority of healers interviewed from the Limpopo Province of South Africa also gained healing power and became healers through inheritance from their parents or grandparents and were men [47–49].

There are people who become healers following episodes of unusual illnesses and thereafter undergo apprenticeship after an experienced healer has received a vision that such a person is fit to become a healer. When the healer accepts this request, then the healer-to-be makes some payments in kind and a ritual for transferring healing power is conducted. If one starts using a medicine for treating others before the ritual of transfer of healing power is conducted, it is believed that the medicine fails to work. In their study, Gessler et al. reported that once a person falls sick and gets instructions from the ancestral spirits to become a healer, he/she has to respond positively to the call because it is considered great disrespect not to take the directives from the ancestral spirits. The result of such disrespect is a series of misfortunes on that person [46]. People who have never had healers in their families or lineage also become healers even if they have not had dreams and visions, nor directives from ancestral spirits. One may choose to undergo apprenticeship and then have a ceremony of transfer of healing power together with an assortment of medicines for managing particular illnesses. This pathway to becoming a healer though rare has also been described by other investigators [46]. In Uganda, traditional healing practices has been officially recognized by the Traditional and Complementary Medicines Act 2019 [40], but the operationalization and integration of traditional health practices into the national health system has not yet been achieved. The process of becoming THPs therefore takes informal cultural processes. In South Africa where THPs are registered and have established associations and training programs, transfer of healing knowledge to their successors follows a rigorous training process that culminates in a graduation ceremony to release a product that is considered safe to the community because the novitiates have been trained and tested, and found worthy to practice as THPs. The novitiates of the Sangomas who have completed training get registered with their own councils before starting independent practices [50]. While in SA the trainees of traditional health practices pay to be taught and initiated into the profession, and as well pay to be shown the necessary medicinal plants [50], the respondents in this study revealed that direct monetary payments to learn the art and practice of traditional healing is prohibited. Indeed, the respondents in this study believe that Luo medicines lose their healing powers when financial transactions are involved during the process of initiation into being a healer, and during healing itself. Healers occupy a unique position in their own societies because of their intimate relationship with the spirit world from whom they get healing powers, and yet they keep close to the people they treat, understanding quite well their daily life experiences so that they can relate with them on equal footing [27]. Among the Acoli, and Luo generally, traditional healing practices are a learned way of being. The

prospective healer learns to be a healer through apprenticeship, whereby the healer shows the medicine to the novice, tells him/her the name of the medicines, the illnesses the medicines treat, spits saliva on the medicines as a blessing, and hands them over to the novice who receives the medicines with both hands, bowing forwards in respect. The novice pays attention but has to say nothing verbally. If he/she says something, he/she forgets the medicine and/or the healing power of the medicine reduces or vanishes completely. Similarly when the novice is sent to harvest a herb, he/she should not look behind after harvesting, otherwise the herb will lose its healing power [27]. The prospective healer learns through observation, listening and participating in the healing practices including harvesting of medicines alongside the healer. The learning is consolidated into the daily experiences of healing. The ethics and conduct of healers are learnt simultaneously with every other aspects of healing including disease symptoms, ways of diagnoses, and treatment options. The healers select people, usually children or grandchildren to whom they pass healing powers and articles of practice including skin of particular animals, calabashes, and cowry shells [28].

We also found that people become THPs through visions and dreams. Most people who become THPs in this way do not have to undergo apprenticeship in healing. Often, these people have been seriously sick themselves. The person does not self-elect to become a traditional healer, but rather become so through the dictates of the ancestral spirits that direct them in the dreams. Some of the respondents in this study reported that they were called to become healers by their ancestral spirits, and that they learned about medicines through dreams and illnesses. Similar findings regarding learning healing practices and medicines that work through dreams have been reported by other investigators [51, 52]. Edwards et al (2009) reported that a diviner is chosen by the ancestors who bestow upon him/her intuitive healing powers. Ancestral calling usually takes the form of powerful ancestral dreams and some unusual encounters [52]. When the individual accepts the call, they become a new person capable of listening to the spirits and then start to mediate the living with the living-dead. Indeed, people become healers through the call of their ancestral spirits directing them to heal their people, give life back to them, and lead them to happiness. After the call to healing by the ancestral spirits, the person has to undergo training, usually through apprenticeship before they can start to provide healing services to the people [51, 53]. So, healing is a gift from the ancestral spirits to a family member who is considered empathic and well-disposed to be responsible over the good health and continuity of life in the clan. If no one develops interests in inheriting the healing practice, the ancestral spirits intervene through dreams and visions, sometimes making a member of the family so very ill, a period during which the spirits reveal to that person the need for carrying on the healing tradition [51]. In South Africa, THPs are viewed as mediators between the ancestors and the people, as counsellors, and as providers of protection at home and on journeys and other expeditions. The THPs would sprinkle or apply medicines that ensures protection and safety in the homestead and or the person on a journey or expedition [54].

In this study, majority of respondents reported that some people who first become possessed by ancestral spirits gain healing powers during a ritual of singing and dancing to their ancestral deities, 'wero jok'. This ritual is often presided over by a diviner. The initial possessions by the ancestral spirits making the person behave as though they are "mentally ill" is considered a calling. The calling often manifests early in adolescence. However, the ritual is never held as early as adolescence so as not to misinterpret a true illness for a calling by the ancestral spirits. The youth called is often observed over time to be sure that indeed the person received a genuine calling and not faking the call for purposes of monetary gain. In northern Kwazulu Natal, a study showed that people are called to become THPs by ancestral spirits. The novice called then undergoes intensive process of learning how healing is conducted and the medicines that are effective for different illnesses [54]. In another study in South Africa, that

involved 50 THPs in five FGDs the THPs perceived traditional healing as an ancestral calling that should be respected and treated as such, basing on cultural norms, not western traditions [55]. The ancestors are central in African metaphysics and philosophy of life. The ancestors are responsible for the good health of the individuals, families and society. The ancestors are offered food in designated shrines by elders during rituals in which members of the clan gather to eat and drink local brews. The elders in charge of the rituals start the party by offering food and drink to the ancestors, as they earnestly ask for good health, reproduction and productivity to young members of the clan, and longevity of the elders. Afflictions in the clan are presented to the ancestors during this ritual. The ancestors usually respond at their own time, for example by calling a member of the clan and revealing to him/her medicines that are effective for treating certain illnesses that have affected the clan [56].

## Perceived causes of illnesses

We found that the Acoli traditional healing system considers the root causes of illnesses as important to any healing processes. This is similar to reports from other communities in Africa, where healers first seek the root causes of ailments before administering any treatments. The African healers offer holistic treatments, aiming at bringing an equilibrium between the physical body, soul, cultural, and spiritual realms [54]. Traditional medicines target the underlying causes of the illnesses, and the success of healing depends majorly on the proper identification of the causes of illnesses. Respondents in this study revealed that illnesses are due to the interactions of several factors including acts of nature and fate of an individual, spiritual forces/attacks, bewitchment, contagions, curses, and transgressions of cultural norms. Earlier work among the Acoli before the devastating LRA/NRM violent conflict showed that illnesses were attributed to spells, curses, ill will of enemies, and evil spirits that attack an individual or community (ies). Illnesses were also attributed to entry of foreign substances into the body, or due to animal and insect bites [57]. Other studies have revealed that a fundamental difference between African traditional medicine and western medicine lies in their concept of the causes of illnesses/diseases and their approach to healing, as well as in the healing methods used. In most African societies the cause of an illness or discomfort is often ascribed to supernatural forces arising from angered ancestral spirits, evil spirits, or the effects of witchcraft. For example, traditional healers in South Africa ascribed the cause of intellectual disabilities to supernatural causes [58]. The African health system therefore tends to delve into the nonmaterial causes of illnesses, unlike biomedicine [46, 59–62].

The majority of respondents in this study attributed illnesses and diseases to fate; that people become sick due to fate and wish of a supernatural being. The respondents said that even if people do everything to protect themselves and do nothing bad, they will still fall sick and die of illnesses if not treated promptly with effective medicines. This view is similar to what earlier studies in other parts of Africa revealed about causes of illnesses. In a study that involved 308 participants, diseases were perceived to be caused by natural agents, supernatural agents, and by combinations of both natural and supernatural agents [59]. Traditional healers in Botswana attributed many symptoms to the will of God [63]. These fatalistic views about causes of illnesses may lead to feelings of helplessness and surrender. It is akin to the missionary teaching of divine omniscience and omnipotence; that God is so very full of wisdom and powerful, and what He decides will always be so. This viewpoints lead to resigning matters to the Will of God, a state of helplessness that can lead to poor health seeking. In fact, the concept of health and illnesses, as well as the treatment of ailments in most societies in the LMICs are intricately related with their religious and spiritual beliefs [64]. In sub Saharan Africa, health interventions that incorporate sociocultural, religious and spiritual beliefs maybe more successful than

those that emphasize only biomedical principles and neglect African traditional medicine [14, 64, 65].

Another cause of illnesses reported by our respondents is attack by vengeance spirits of the dead who died in states of unhappiness, distress and perceived neglect by the members of their families and or communities that should have provided them protection and support. Respondents believed that the dead in the world of the living-dead can take revenge on specific persons, or families or sometimes the community at large if they perceived that they were not treated with dignity at the time of their deaths. The illnesses caused by spirits attacks include mental disorders, wondering away from home, walking aimlessly all day, and inappropriate behaviors. Some of the illnesses due to spirits attacks include epidemics of strange diseases that affect the whole community. An example of such disorders in the Acoli region include nodding syndrome, a disease that affects children aged 5–18 years and causes marked physical and mental retardations. The people in the affected regions attributed the causes of nodding syndrome to several factors including spirits attacks by vengeance spirits of people who were brutally killed, did not get decent burials, and whose bones were all over the place and on which children stepped and played with to the annoyance of the spirits [66–68]. Among Acoli and most African societies, illnesses are often caused not just by what, but also who. Connivance between the spirit world through some supernatural forces and earthly factors are considered responsible for illnesses in most African societies [69]. The social circumstance of the life of the affected person and his/her society is critical to explaining illness occurrences. The Acoli believe that illnesses do not occur randomly; even fate has a reason to befall the affected person at the time it does. The belief system of a people tends to shape the people's outlook of illness causations. In most African communities, the concept of spirits possessions is common and associated with certain symptoms. The person who is possessed with spirits often behave in manners that show they are not in control of their faculties, but some supernatural forces or gods are in charge. They display features of a powerful person with powers above that of ordinary humankind and yet they are known to be humankind whose parents and siblings are normal humankind living in the same society [70]. The Acoli sub region had a long period of spiritual possession campaign by both Alice Lakwena Holy Spirits Movement and the Joseph Kony Lord's Resistant Army Movement (LRA/M). These movements instilled it into the minds of the Acoli, emphasizing earlier beliefs that spirits can possess a person and make them do marvelous things, including repulsing bullets from guns and artilleries. People in the region may therefore continue to explain certain unusual occurrences and abnormal human behaviors using the spirit possessions theory.

Curses are thought to cause illnesses through causing imbalances in the affected person/ people, creating an internal state of restlessness in association with physical symptoms e.g. swelling of the abdomen that does not respond to usual treatments. If young people behave in manners that anger the elders and the living-dead, and consistently do so even when repeatedly cautioned, the elders can curse that person and he/she wonders around and eventually develops scandalizing physical symptoms. The curse can follow the person wherever he/she goes, and cause him/her misfortunes in their lives. A recent study among the Luo of western Kenya showed that some illnesses are a consequence of misadventures and or conducts that are contrary to established cultural norms [71]. Our findings are also similar to reports from other studies among traditional healers from Africa in which the healers reported that ancestral displeasure due to disobedience of the young people lead to curses and illnesses. Even infertility, hydrocele and hunchback (TB of the back bones) have been associated with curses and misfortunes arising from angering elders and the living-dead, and antisocial behavior and disregard to cultural norms and values [69, 72, 73]. Healers in Botswana perceived that antisocial behaviors including sexual immorality that have become more prevalent in the post-

colonial era was responsible for much of the illnesses that have afflicted the African populations. The healers conceived that blood mix when two people engage in sexual intercourse, and the mixture is normal if the two people do not again mix it with another person's blood. However, if a person who has already mixed their blood again mixes blood with another person, the product becomes toxic and leads to illnesses [63]. To the Africans, the living have to be in harmony with the ancestors. If some people have repeatedly engaged in conducts contrary to what the culture dictates, they may invite ancestral annoyance which usually present as illnesses, often causing swelling of the abdomen or disorders of the mind. Some diseases therefore are thought to be a result of a disequilibrium between humans and their ancestral spirits [56, 73].

In this study, respondents acknowledged that malevolent attacks by sending evil spirits, witchcraft and sorcery are yet other ways by which people become ill. Most of the illnesses acquired through these sources are strange. Witches and sorcerers are thought to have some power that they can send onto people, leading them to fall sick or develop some disorders. In addition, the respondents reported that some illnesses such as dysmenorrhea in young ladies are passed on by a woman who is suffering from the same disorder when she touches the waist of a lady who does not have the disorder. It is also thought that dysmenorrhea passes on to a woman if she uses cleaning materials prepared and used for cleaning saucepans or water calabashes by a woman who has the disorder. Our finding coheres with results from other studies that humankind can send illnesses through some extra-sensory means to a person far away from the sender [69]. A study among the Luo of western Kenya showed that evil attacks or revenge by sending evil forces are common causes of ill health [71]. A study in Malaysia also showed that sorcery and sending evil spirits to attack people are a common cause of ill health [74]. Among 100 THPs in Tanzania, up to 30% thought epilepsy is an illness caused by witchcraft [75]. Some illnesses perceived to be due to spirits attacks affect babies or children who are born with some physical abnormalities e.g. being born with teeth, beards, disproportionately big heads (hydrocephalous), or growth and intellectual abnormalities. Most often such children are perceived to cause misfortunes or other problems to the family or community at large. So, most such children are often killed as soon as they have been recognized as spirit children [76, 77]. In most African societies, mothers of children with severe intellectual disabilities or physical malformations/deformities often get stigmatized and discriminated against by members of their societies [78–81].

A substantial minority of respondents also reported that some diseases run in families and clans, and therefore may have genetic causes. The illnesses perceived as due to genetic disorders have been associated with strange physical defects or disabilities and or mental disorders or intellectual disabilities. These disorders and defects have serious social significance among the Acoli people in that elders always tell their sons and daughters never to sexually relate with and or marry people from families with disorders that are thought to have genetic causes. The prohibition against marriages from certain families also applies for illnesses that are thought to be due to curses and misfortunes because those disorders will ultimately affect the offspring of their sons or daughters. When sons and daughters are being prohibited from marrying from such families, the society is not just stigmatizing and discriminating against such persons, but rather making attempts to ensure that such illnesses are prevented by never allowing them any chances to propagate themselves but rather die with whoever already has them. In South Africa, a study on grandmothers' perceptions about children born with genetic disorders–disabilities and abnormal features on the child, revealed that causes of such inheritable disorders are many and relate to the conduct of the woman both before and during pregnancy, advanced maternal age, as well as engaging in culturally prohibited activities including sexual intercourse between blood relations. A consensus approach to the prevention of such genetic disorders

included killing of the abnormal children to avoid them passing their genes on, and providing cleansing rituals for the mother before they get pregnant again [82]. In Kenya, an in-depth interview with 13 families affected with sickle cell disease (SCD) revealed that mothers of children with SCD are generally stigmatized for having children with genetic disorders [83]. The stigma and burden of care of the affected children affect these women and they often withdraw from social lives and relations that could lead to having any more children, curtailing their reproductive potential an indirectly preventing passing on the genetic disorders. In Zimbabwe, up to 19% of mothers with cleft lips and palate were discriminated against and divorced mainly in relation to the abnormalities in their children. Serious physical abnormalities like cleft lips and palate are associated with promiscuities by the mothers during pregnancy [84].

## Diagnostic approaches

The respondents in this study expressed the need for a correct diagnosis of any illness before starting treatment. They emphasized that their medicines are for specific category of illnesses, and therefore knowing the nature and cause of an illness is key to successful treatment. The healers used different approaches to diagnoses depending on the presentation of the patients. The approaches used in diagnoses include listening carefully to the symptoms and signs of illnesses, observing patterns of occurrences and events during the onset of the illness, use of animals/birds and objects, consultations with ancestral spirits, performing diagnostic rituals, and sending patients to do laboratory tests in the health facilities. Respondents said they would first consult amongst themselves before sending patients to health facilities. Whenever they fail to come up with a diagnosis, when they are not familiar with a particular illness presentations, then they would send such a patient to the biomedical facilities for diagnosis and treatment. They often asked such patients to return and inform them what the biomedical professionals would have decided and done regarding the illnesses. These different approaches to diagnosis have been described by scholars in African indigenous medical knowledge, and are often undertaken in series [46, 85]. A study in four African countries showed that THPs use multiple techniques including just looking at the patient, taking comprehensive history and performing thorough examinations, consultations with the ancestral spirits, and dreams and visions [86]. Among the African traditional medicine practitioners, diagnosis of illnesses is both an art and science based on methods that have been learnt through apprenticeship and rituals. African THPs seek to discover the ultimate causes of illnesses—seeking for the immediate causes of the illness as well as why the illness occurred in that person at the particular time. This comprehensive process includes looking out for who may have been responsible for the cause of the disease [87]. Similar findings were reported in SA, where THPs used a number of approaches to diagnose the causes of illnesses including consultations with the ancestral spirits, throwing of certain diagnostic objects including bones and specially cut shoes, and then reading the signs based on how the object fell, the direction it faces and whether or not it has fallen on some other object of significance. The ancestors are believed to talk to the healer through the diagnostic objects. The medicine man/woman (diviner) has to report only that cause revealed to him/her by the spirits during the incantation. He/she should never tell lies regarding the revealed cause of an illness, otherwise, he/she suffers retributions of the spirits [56]. Some THPs dream about the causes of illnesses while in their sleep following interaction with the patients, while other THPs use certain rituals including dressing in beads and or specific clothing and then washing the patients with herbs or drumming [53].

One the most common approaches used by the THPs to diagnose illnesses is listening to the history of the illnesses, evaluating the symptoms and signs, and then interpreting them within the context of other significant life events of the people during the season of the year.

The circumstances and the life of the person and other factors around the occasion when the person falls sick greatly contribute to the preferred diagnoses. When a certain set of symptoms occur in many people in the community at the same time, the healers always focus on the predominant socio-cultural and political circumstance of the life of the community when seeking to arrive at a diagnosis. The reason why the symptoms have occurred in such large numbers tend to signal the preferred diagnosis which may be different from when the same set of symptoms and signs occurred in the same community but affecting only few people. Similar findings have been reported in studies in other parts of Africa. An in-depth interview study with THPs in SA revealed that healers use a common approach to healing, which involves deep interrogations in the spiritual, physical, psychological and social spheres of life of the sick person in order to uncover any imbalance that could be responsible for the ill health. The THPs then administer herbal preparations or specific healing rituals depending on the discovered cause of the illness [53]. In Ghana, THPs reported they more commonly use history of the illness and physical examinations of the patients including their hygiene status to diagnose mental illness [88]. Psychological disturbance, social conflicts and or spiritual attacks are perceived causes of disequilibrium that lead to physical and or mental illnesses [89]. Healing therefore according to African cosmology requires re-establishing this equilibrium. The healer explores what the patient has given to life, and what life has given to him/her, considering all the misfortunes and circumstances that could have resulted in unexpected life challenges, and then determining how to restore the equilibrium.

Spirit consultations and divination during diagnostic rituals are a common diagnostic process used when diagnoses are challenging to the healers. When the signs and symptoms, or history of illnesses are not leading the healer to a particular illness or disease, then the healers consult the spirits, either directly through use of some items or through the diviners as medium of communications with the spirit world. People who suffered from unusual, strange and repetitive illnesses consulted the spirits of their ancestors through the diviners as medium of communications. Misfortunes were also diagnosed through spirits consultations and divination during diagnostic rituals. Spirits consultations have been used in diagnosis of several illnesses including snakebites [46, 90]. During the divination process, the spirits also inform the healers of the appropriate medicines for the illness. Diagnosis in the African traditional healing process is an elaborate process that may require a diagnostician, usually a diviner who seeks to establish: (i) the immediate/proximate causes of illnesses or the natural causes that are agents of the supernatural forces, (ii) the primary or underlying cause of the disease i.e. the reasons responsible for or forces/person that approved of the illness, (iii) the reasons for the actions of the supernatural in causing or approving of the illness occurrence, (iv) the necessary sacrifices or amends to be done so that the medicines work on the illness, and (v) the other healers and items of healing required for the performance of the actual the ritual sacrifice and the practical healing using the identified medicines. The question of who is responsible for the cause of illness is considered very important in the diagnostic process [69, 91, 92]. The respondents in our study said they engage in such elaborate divination process described by Okwu et al. [69] only when the disease is strange, seems to have spirit component, and difficult to understand. Not all healers engage in divination or consult with the ancestral spirits during the process of diagnosis of illnesses. Divination is a highly prevalent and revered diagnostic approach among traditional healers in most parts of Africa [93]. The manner in which someone gained healing power and became a healer determines their divination ability. Most healers who had never had healing history but became sick and then developed interest in healing, and thereby had healing power transferred to them through a ritual often do not gain divination power. Healers who inherited healing power from parents or grandparents without divination power also do not get that power. These categories of healers refer their patients who need spirit intervention

through divination to healers with such powers. Majority of healers do not start any treatment until when they arrive at a diagnosis based on their preferred diagnostic approaches. The patients sometimes visit the healers several times before a diagnosis is reached. When the healer fails to arrive at a diagnosis, he/she refers to another healer or to allopathic medicine [53].

Key signs that point towards the reason for illness include birth defects which are usually considered a sign that the spirits of the rivers or gods of the rivers (*Jok Kulo*) were annoyed either with the parents, the nature or circumstance of the pregnancy including where in space the sex that led to the pregnancy occurred and or annoyance with the clan of the child born with such defects. If the defects are severe and considered incompatible with life, the healer advises the parents of the child to take and offer the child to the god of the river. Once the family has deliberated on the same and agreed to the guidance of the healer, then the mother of the child loosely straps the child on her back, enters into the deep part of the river then releases the strappings so that the child drowns in the river. While the child is drowning, the mother pretends to try and save the child, but does not really reach out to hold up the child who eventually drowns away. A ceremony to mourn the child is eventually organized with a few elders who talk with the god of the river, saying among others "please receive the child that we have given back to you; it was yours that is why you made it born in that way; who competes with the owner; you are the most powerful who owns the universe; receive the child back to; give luck and peace to the family; let our daughter now give birth to another child who will now be ours. And all other elders agree to the incantation unanimously "eyoo" (meaning, yes). Similar findings have been reported where children born with severe malformations diagnosed as action of the water spirits. These children are usually expected to die early because the water spirits needs them. Therefore, the parents do not seek further medical attention so that the child may die as soon as possible and goes to the water spirits before the spirits get annoyed [92]. In a sense, once the diagnosis is arrived at, the effective treatment is also prescribed. In the circumstance of severe congenital malformation, the treatment is no treatment so that the child dies and goes to the gods of the river who need the child for some greater purposes than the people on earth needed the child. So among the Acoli, as it is in most African cultures, the preferred treatment for a diagnosed illness or disorder is not just directed at symptoms or the disease process, but also the reason the illnesses started. The treatment regime may involve use of several different herbs and or approaches including incantation, but all these are decided at the time of the diagnosis. The various treatment approaches may aim at different components–the symptoms, the disease process as well as the reasons underlying the disease evolution. Sometimes herbs are given for the disease symptoms while another approach is used to deal with the problems that have caused the disorder [46].

We found that use of certain objects and materials for diagnoses is a common practice among the healers, especially when the cause of the illness is not obvious from the presenting symptoms and signs. The healers often throw up some objects and observe the way they land on their diagnostic surface, often skin of animals used during when they were being initiated into healer process. Often cowry shells (*gaggi*) and special (diagnostic) shoes are thrown up after shaking several times as the healer utters some words in low tone, in essence talking to the ancestral spirits. He/she keenly observes the manner in which the object settles down, in particular the direction a certain pre-marked part faces (up, down, towards the door or away) when thrown up, for example particular kind of shoes are thrown up and the direction the front points tells of certain particular illnesses. Some healers use particular bones from specific animals. They throw up the bones and the manner in which the bones land on the skin spread on the floor, and the direction of a certain pre-identified tip tells the diagnosis. Often times the bones are either passed on from their predecessors or parts of the bones got from the animal/

chicken slaughter during the ritual for transfer of healing power to the THP during initiation into the healing profession. When the diagnosis of a rather rare illness becomes challenging and the illness becomes severe in spite of treatments, the healers go for the gut signs or tests, which are done by opening up a goat or chicken to see the diagnosis on the intestines/gut. There are particular healers who know how to read the diagnosis. The gut sign is also used in circumstances when the patient presents with uncommon symptoms or presents in a manner that is not typical of any known or common illnesses. These diagnostic procedures have been described in studies among other communities in Africa. Some healers diagnose spirits attacks when they open the gut of a chicken or goat and finds some parts of the guts missing, evidence that an evil spirit attacked and extracted parts of the gut. It is spirit attack because normal mortal being cannot enter and extract portion of guts from a living animal/bird. Other healers identify certain swellings or color changes on the intestines and attribute them to particular illnesses. Although we asked about the diagnostic procedures, we did not insist to gain detail knowledge on the specific ways by which the healers really arrive at the diagnosis based on the gut tests or throwing of bones and or other materials because they considered these their trade and would not wish to disclose the details. The chicken or goat that are opened up to assess for cause of illnesses are often brought by the patients. Often, the sex of the animal matches that of the patient. Similar results have been reported from studies in Tanzania [46]. The use of divination bones got from both domestic and wild animals and birds, and as well as using other plant materials for diagnosis of illnesses is prevalent in many other communities [48, 94]. The throwing of the divination bones form a central role in the diagnosis especially of severe illnesses that do not respond immediately to treatments. The healer says certain figurative expressions and praises while throwing the bones, and calling upon the ancestral spirits to intervene and direct the manner in which the bones work to show the diagnosis in the menacing illnesses [95]. The healers use these natural items in diagnoses because of the perceived closeness between humankind and nature including the physical environment. The Africans believe that nature and the environment in general exist to support, nurture and ensure the wellbeing of humankind and therefore should support any processes that aim to better and reinstate the health of humankind [51]. Throwing of diagnostic bones was reported as a main way THPs in South Africa diagnose cervical cancer [96].

When the traditional approaches to diagnoses do not yield satisfactory results, and or when the healer thinks the illness is of a modern nature, he/she refers the patients to the biomedical facilities. The modern or *muzungu* (Referring to a white man) illnesses are disorders that THPs and African societies consider recent and only recognized during the colonial and postcolonial eras. These modern illnesses include cancers, hypertension, and diabetes. They are best diagnosed in the health facilities using the instruments of the white man. Referral of patients to biomedical facilities was also reported by THPs regarding diagnosis and management of infertility. Whenever their diagnostic approaches do not reveal the cause and reasons for the infertility, the healers would refer to the health facilities because the cause maybe biomedical [72].

## Limitations

Transferability of findings from this study is potentially restricted to situations where the cultures are reasonably similar to the Luo cultures because traditional healing practices are so very culture bound. Recruiting respondents from both East and West Acoli increased the chances of appropriate transfer of results to other Luo communities and related societies. Second, the details of the rituals for transfer of healing powers and diagnostic processes have not been captured because the THPs were concerned with theft of their techniques (intellectual

property rights) without any benefits to them; so we did not observe the processes of diagnosis and healing rituals. We recommend case studies and respondents' observational approaches to allow for detail documentations of the diagnostic and healing ritual processes. The healers require compensations before disclosing the details of their practices. Thirdly, respondents' validation was not conducted with all the 22 respondents because of logistical reasons and time constraints. Nevertheless, we conducted eight respondents check. We did not find any significant additional information nor disagreement with prior information. We therefore believe that the data presented here represents the views of the respondents.

## Conclusion

The Central Luo in Northern Uganda have shared customs regarding acquisitions of healing power and entry into the healing vocation. Healing is considered a calling by one's ancestors resting in the world of the living-dead who require their children and grandchildren to flourish and multiply uninterrupted by illnesses. The healers are aware and uphold perceptions regarding causes of illnesses and treat patients on the basis of the perceived causes. Unfamiliar illnesses are often diagnosed through consultations with the spirits and through rituals. If an illness is considered to be associated with modernization, then the healers refer such a patient for diagnoses and treatment in the biomedical facilities. Findings on how people gain healing powers and become healers can inform policies on registration and licensing of traditional health practitioners. Regulatory bodies concern with THP registration can screen and exclude imposters who have not undergone the known processes of becoming a healer. Knowledge gained from this study has potential to inform training of biomedical health professionals on critical aspects of traditional health practices to allow for meaningful integrations of biomedical and traditional health practices.

## Acknowledgments

The authors appreciate Professor Fiona M. Walter and the Cambridge-Africa Research Program for the funding that enable the study conduct. ADM appreciates Professor Fiona M. Walter, Professor Henry Wabinga, Professor Elialilia S. Okello, Professor Elizeus Rutebemberwa and Professor Sunita Vohra for their mentorship. The authors appreciate Ojara Francis who participated in the recruitment and data collection. We are also grateful to the respondents for their active participation.

## Author Contributions

**Conceptualization:** Amos Deogratius Mwaka, Christopher Garimoi Orach.

**Data curation:** Amos Deogratius Mwaka, Jennifer Achan.

**Formal analysis:** Amos Deogratius Mwaka.

**Funding acquisition:** Amos Deogratius Mwaka.

**Investigation:** Amos Deogratius Mwaka, Jennifer Achan, Christopher Garimoi Orach.

**Methodology:** Amos Deogratius Mwaka, Christopher Garimoi Orach.

**Project administration:** Amos Deogratius Mwaka, Jennifer Achan.

**Resources:** Amos Deogratius Mwaka.

**Software:** Amos Deogratius Mwaka.

**Supervision:** Amos Deogratius Mwaka, Jennifer Achan.

**Validation:** Amos Deogratius Mwaka, Christopher Garimoi Orach.

**Visualization:** Amos Deogratius Mwaka, Christopher Garimoi Orach.

**Writing – original draft:** Amos Deogratius Mwaka.

**Writing – review & editing:** Amos Deogratius Mwaka, Jennifer Achan, Christopher Garimoi Orach.

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
