## [Decision Letter · Decision Letter 0]

1 Dec 2022

PONE-D-22-24830Traditional health practices among the Acoli in northern Uganda: An exploration of the views of traditional health practitioners on becoming a healer, perceived causes of illnesses, and diagnostic approachesPLOS ONE

Dear Dr. Mwaka,

Thank you for submitting your manuscript to PLOS ONE. After careful consideration, we feel that it has merit but does not fully meet PLOS ONE’s publication criteria as it currently stands. Therefore, we invite you to submit a revised version of the manuscript that addresses the points raised during the review process.

We look forward to receiving your revised manuscript.

Kind regards,

Adetayo Olorunlana, Ph.D.

Academic Editor

PLOS ONE

Journal Requirements:

Reviewers' comments:

Reviewer's Responses to Questions

**Comments to the Author**

1. Is the manuscript technically sound, and do the data support the conclusions?

Reviewer #1: Yes

Reviewer #2: Yes

2. Has the statistical analysis been performed appropriately and rigorously? 

Reviewer #1: Yes

Reviewer #2: N/A

3. Have the authors made all data underlying the findings in their manuscript fully available?

Reviewer #1: No

Reviewer #2: No

4. Is the manuscript presented in an intelligible fashion and written in standard English?

Reviewer #1: Yes

Reviewer #2: Yes

5. Review Comments to the Author

Reviewer #1: Thank you for this interesting paper. It is quite detailed and explores different areas. There are minor points that need to be clarified.

1. Traditional Healers describe a broad category of healers including herbalists. It would be useful if you could specify the categories of the participants included in your study. You mention that you used the list of registered THPs from the government offices. Are they registered according to their specialties? There is a reference to witch doctors by the study participants in the results section and witches and sorcerers are mentioned later in the paper. It would help if these were defined or described briefly. Could you also clarify if witch doctors would be registered as practitioners since you also mention in the discussion that they play a vital role in the diagnosis or divination before appropriate intervention is decided on?

2. You have highlighted one harmful practice regarding the killing of children with malformations. Are there any other specific harmful practices that should be discouraged? These could be highlighted, and specific recommendations made to minimise the practice.

3. Lines 757-760 should be rewritten for better clarity on how the female healer is expected to conduct herself.

4. Does the "nodding syndrome" mentioned in line 1017 have an equivalent bio-medical diagnosis?

5. Typos noted:

The reference in line 1105 should conform to the numerical format

Under ethical consideration, the sentence starting "Privacy was achieved .... " seems to have a word missing

Reviewer #2: Thank you for the opportunity to review this paper which provides a thick description of the becoming a healer, perceived causes of illness and diagnostic approaches of traditional healers in Uganda.

My overall impression of the paper was that it was very long and there was a lot of repetition most notably between the results and discussion. Below I provide some specific comments, questions and suggestions that I hope will strengthen the manuscript.

1. A very interesting and important issue raised in the Introduction, partially as a justification of why this study is needed, is that a large number of people from Central Luo have been displaced especially between 1987-2006, and this quite likely impact traditional healing practices. However, this topic does not appear to be referred to again (i.e., after the introduction). It seems like a very important question to delve into given that the finding that a key pathway to becoming a healer was a transfer of knowledge from parents, grandparents or senior healers - all of which was quite likely significantly disrupted during the massive displacement that occurred.

2. Study setting - it would be helpful to know if/how services of traditional healers is funded in Uganda

3. Study Design/data collection - the study is described as "an ethnographic study that used in-depth interviews and observations of the practices and memorabilia of the traditional healers." However the data collection and analysis only describes data collection and analysis of the interviews. Observation is such an important part of an ethnography that this leads me to believe that either this was not actually an ethnographic study design or that this part of the methods needs to be added including a description of the observation techniques, time in the field, and analysis of the observations.

4. I would recommend that the results -- the themes and subthemes -- and the discussion be combined. There is a currently a lot of overlap and at the same time I found many instances of new information (i.e., results) in the discussion that was not in the results section. This could be remedied by combining the sections. This could also be used as way to decrease the length of the paper.

5. As noted above, I was disappointed that there was no mention or query of any changes as a result of the massive displacement described in the introduction. The findings were presented without any sense of context -- has it always been this way? what, if anything, has changed over time and specifically related to the displacement?

5. The current findings and discussion sections are very descriptive. It was very interesting to read about these traditional healing practices, but it is not clear to me who the intended readership is and how this knowledge might be applied/useful. For example, could the authors identify how this knowledge might be used to help ensure adequate health care access for the population? or to make changes in the health care system or for some other purpose. As currently written, the information is interesting, but the "so what?" is not clear.

6. The boxes at the end of the paper appear to contain additional quotes -- some explanation of why they are included, how these quotes were chosen and/or the purpose of the putting this information in boxes (other than to try to reduce the word count of the paper) would have been helpful.

6. PLOS authors have the option to publish the peer review history of their article (what does this mean?). If published, this will include your full peer review and any attached files.

Reviewer #1: **Yes: **Prof. Caleb Joseph Othieno

Reviewer #2: No

---

## [Author Response · Author response to Decision Letter 0]

4 Feb 2023

4th February 2023

Point by point responses to the Editorial and reviewers’ comments to our Manuscript, ID number: PONE-D-22-24830, Titled: “Traditional health practices: a qualitative inquiry among traditional health practitioners in northern Uganda on becoming a healer, perceived causes of illnesses, and diagnostic approaches”

Dear Editor in Chief and the Reviewers,

We are appreciative and extremely grateful for the time, efforts and knowledge/skills you have committed to guide us to improve the quality and readability of our manuscript. We have addressed the comments as follows:

Reviewer #1: 

Thank you for this interesting paper. It is quite detailed and explores different areas. There are minor points that need to be clarified.

1. Traditional Healers describe a broad category of healers including herbalists. It would be useful if you could specify the categories of the participants included in your study. You mention that you used the list of registered THPs from the government offices. Are they registered according to their specialties? There is a reference to witch doctors by the study participants in the results section and witches and sorcerers are mentioned later in the paper. It would help if these were defined or described briefly. Could you also clarify if witch doctors would be registered as practitioners since you also mention in the discussion that they play a vital role in the diagnosis or divination before appropriate intervention is decided on?

Very true that is. We have limited this study to THPs who are herbalists. We have included this under study population section of the methods.

The THPS are not registered according to their specialties; the register includes herbalists, those who conduct healing rituals, and use prayers to heal (spiritualists), as well as the diviners or witchdoctors.

Witchdoctors are also registered but not as witchdoctors; they are registered as healers because they usually provide some medicinal products in addition to their psychosocial prescriptions. However when we reached the healers, we inquired more about their practices and included only those that predominantly use herbs.

When the herbalists get challenges in diagnosis or believe that the particular illness require intervention of a diviner (also derogatively referred to as witchdoctor), they refer the patients to them. That is the context we refer to in the results and discussions sections of our manuscript. Generally, the term witchdoctor is derogative and used in a negative sense. The practitioners themselves would describe their trade as diviners. But we have chosen to stay clear of the linguistic but stick with the concept of divination and incantation.

2. You have highlighted one harmful practice regarding the killing of children with malformations. Are there any other specific harmful practices that should be discouraged? These could be highlighted, and specific recommendations made to minimise the practice.

We did not inquire for other potentially or actually harmful practices beyond what were volunteered by the respondents. Not because we did not consider that as important but rather because we wished to focus our findings. So we did not explore much into details regarding this practices. The THPs are often very excited to be interviewed by biomedical practitioners or their associates and will always talk about everything and can potentially lose focus or make the transcripts unnecessarily long and difficult to tease out.

For sure, the debate on killing of infants with physical deformities or observed mental deficiencies are not uncommon in the LMICs especially sub Saharan Africa. It is an excellent area within the context of traditional medicine practices for further explorations. 

3. Lines 757-760 should be rewritten for better clarity on how the female healer is expected to conduct herself.

Thank you for the concern. This has been revised out in the main text because we did not find much supporting evidence from results. We are sorry to have initially emphasized it, mainly from our own experiences in the study community.

4. Does the "nodding syndrome" mentioned in line 1017 have an equivalent bio-medical diagnosis?

Nodding syndrome is actually the bio-medical diagnosis. The locals call it “lucluc”, a descriptive term referring to the predominant symptoms that precedes generalized convulsive disorders. 

For your reading pleasure, you may check out these articles on nodding syndrome: 

1. DEOGRATIUS, M. A., DAVID, K. L. & CHRISTOPHER, O. G. 2016. The enigmatic nodding syndrome outbreak in northern Uganda: an analysis of the disease burden and national response strategies. Health policy and planning, 31, 285-292.; 2. IDRO, R., OPOKA, R. O., AANYU, H. T., KAKOOZA-MWESIGE, A., PILOYA-WERE, T., NAMUSOKE, H., MUSOKE, S. B., NALUGYA, J., BANGIRANA, P. & MWAKA, A. D. 2013. Nodding syndrome in Ugandan children—clinical features, brain imaging and complications: a case series. BMJ open, 3, e002540.; 3. MWAKA, A. D., SEMAKULA, J. R., ABBO, C. & IDRO, R. 2018. Nodding syndrome: recent insights into etiology, pathophysiology, and treatment. Research and reports in tropical medicine, 9, 89.; 4. POLLANEN, M. S., ONZIVUA, S., ROBERTSON, J., MCKEEVER, P. M., OLAWA, F., KITARA, D. L. & FONG, A. 2018. Nodding syndrome in Uganda is a tauopathy. Acta Neuropathologica, 136, 691-697.

5. Typos noted: 

The reference in line 1105 should conform to the numerical format.

This is revised; thanks.

Under ethical consideration, the sentence starting "Privacy was achieved ...." seems to have a word missing.

There was error in editing, and sentences were repeated in parts. This has been revised to bring out aspects of both privacy and confidentiality. Thanks for the keen observation and guidance.

Reviewer #2: 

Thank you for the opportunity to review this paper which provides a thick description of the becoming a healer, perceived causes of illness and diagnostic approaches of traditional healers in Uganda.

My overall impression of the paper was that it was very long and there was a lot of repetition most notably between the results and discussion. Below I provide some specific comments, questions and suggestions that I hope will strengthen the manuscript.

We are glad you enjoyed reading the paper.

We take serious note of your comment on the length and repetitions in the paper. We have therefore conducted a revision to crystalize the piece of work and reduce the length without losing content.

1. A very interesting and important issue raised in the Introduction, partially as a justification of why this study is needed, is that a large number of people from Central Luo have been displaced especially between 1987-2006, and this quite likely impact traditional healing practices. However, this topic does not appear to be referred to again (i.e., after the introduction). It seems like a very important question to delve into given that the finding that a key pathway to becoming a healer was a transfer of knowledge from parents, grandparents or senior healers - all of which was quite likely significantly disrupted during the massive displacement that occurred.

We agree entirely with your concerns. We have distilled this aspects in the discussion section of the revised main text. While we would have loved to compare the perception before and after the conflict, the data is limited on this aspects and so our thick description of the current status serves to set stage for the post conflict status quo. We believe that future review studies could attempt to compare and evaluate the impact of the conflict on traditional health practices. The scope of this paper does not allow us to delve deep into the comparative analysis without losing track and abandoning our data.

2. Study setting - it would be helpful to know if/how services of traditional healers is funded in Uganda

Thank you for this legitimate concern. Traditional health practices have not been recognized and accepted national in Uganda, though widely practice. There is limited published data to this effect. In our experience, the healers fund their services from their earnings. 

3. Study Design/data collection - the study is described as "an ethnographic study that used in-depth interviews and observations of the practices and memorabilia of the traditional healers." However the data collection and analysis only describes data collection and analysis of the interviews. Observation is such an important part of an ethnography that this leads me to believe that either this was not actually an ethnographic study design or that this part of the methods needs to be added including a description of the observation techniques, time in the field, and analysis of the observations.

We agree that the ethnographic component is not well described to suitably fit an ethnographic study design. The authors are all indigenous Acoli/Luo who have lived their lives in the region. However, we specific regard to this study, no much time was specifically dedicated to observe the practices beyond the interview time. We have therefore preferred, on the strength of your guidance to revise the design as “a qualitative interview study design”.

4. I would recommend that the results -- the themes and subthemes -- and the discussion be combined. There is a currently a lot of overlap and at the same time I found many instances of new information (i.e., results) in the discussion that was not in the results section. This could be remedied by combining the sections. This could also be used as way to decrease the length of the paper.

Thank you for the recommendation of combining the results and discussions sections. However, to avoid loss of suitable data in the results and not mix our own interpretations with voices of the respondents, we have preferred to revise the results and discussions sections – removing our own thoughts from the results, as well as removing new results from the discussion section. We have left the structure to the usual Journal’s requirements with separate results and discussion sections.

We have revised the manuscript and cut down on the length too.

5. As noted above, I was disappointed that there was no mention or query of any changes as a result of the massive displacement described in the introduction. The findings were presented without any sense of context -- has it always been this way? What, if anything, has changed over time and specifically related to the displacement?

We agree with your concerns. However, we did not seek to compare and contrast practices before and after the conflict and or evaluate the impact of the conflict on cultural practices including healing. Our data does not expressly bring out issues related to changes in the practices although it is expected that the conflict could have significantly changed the ways – but maybe it did not. A study directed to synthesize existing published data will expose the extent to which the conflict may have impacted traditional health perceptions and perhaps practices.

5. The current findings and discussion sections are very descriptive. It was very interesting to read about these traditional healing practices, but it is not clear to me who the intended readership is and how this knowledge might be applied/useful. For example, could the authors identify how this knowledge might be used to help ensure adequate health care access for the population? Or to make changes in the health care system or for some other purpose. As currently written, the information is interesting, but the "so what?" is not clear.

We are glad you enjoyed reading the work. Readers include academics, clinicians and health policymakers. Then the “so what, if all these and that are as said”: First we needed to expose the wealth of knowledge on how one becomes a practitioner, the perceptions of the traditional health practitioners on causes of illnesses, and how diagnoses are made. We believe we have well achieved the exposition as per your assessment. Second, healthcare professionals and the public can therefore now understand the perspectives of the THPs, and this understanding potentially modifies the relationship and cooperation between the practitioners of the biomedical and traditional health systems. Third, knowledge of how people become practitioners can guide the government and other stakeholders on registration and licensing of THPs because it will be easy to trace the authenticity of one’s claim of being a THP, thereby minimizing false traditional health practices by imposters. Safety of the population will be improved.

We have distilled these ideas into the discussions and conclusion. Thanks for the guidance.

6. The boxes at the end of the paper appear to contain additional quotes -- some explanation of why they are included, how these quotes were chosen and/or the purpose of the putting this information in boxes (other than to try to reduce the word count of the paper) would have been helpful.

These have been added as additional information on the voices of the respondents. Technically enriching the discourse without unnecessarily lengthening the paper. This has been referred to in the results section.

Kind regards

Amos Deogratius Mwaka

Corresponding author

---

## [Editor Report · Decision Letter 1]

16 Feb 2023

Traditional health practices: a qualitative inquiry among traditional health practitioners in northern Uganda on becoming a healer, perceived causes of illnesses, and diagnostic approaches

PONE-D-22-24830R1

Dear Dr. Mwaka,

We’re pleased to inform you that your manuscript has been judged scientifically suitable for publication and will be formally accepted for publication once it meets all outstanding technical requirements.

Kind regards,

Adetayo Olorunlana, Ph.D.

Academic Editor

PLOS ONE
---

## [Editor Report · Acceptance letter]

21 Feb 2023

PONE-D-22-24830R1 

Traditional health practices: a qualitative inquiry among traditional health practitioners in northern Uganda on becoming a healer, perceived causes of illnesses, and diagnostic approaches 

Dear Dr. Mwaka:

I'm pleased to inform you that your manuscript has been deemed suitable for publication in PLOS ONE. Congratulations! Your manuscript is now with our production department. 

Kind regards, 

on behalf of

Associate Professor Adetayo Olorunlana 

Academic Editor

PLOS ONE